

# Recovery of deep-sea meiofauna community in Kaikōura Canyon following an earthquake-triggered turbidity flow

Katharine T. Bigham[1,2], Daniel Leduc[2], Ashley A. Rowden[1,2], David A. Bowden[2], Scott D. Nodder[2] and Alan R. Orpin[2]

[1] School of Biological Sciences, Victoria University of Wellington, Wellington, New Zealand
[2] National Institute of Water and Atmospheric Research, Wellington, New Zealand

Corresponding author
Katharine T. Bigham,
katie.bigham@niwa.co.nz,
bighamkt@gmail.com

## ABSTRACT

Turbidity flows can transport massive amounts of sediment across large distances with dramatic, long-lasting impacts on deep-sea benthic communities. The 2016 $M_w$ 7.8 Kaikōura Earthquake triggered a canyon-flushing event in Kaikōura Canyon, New Zealand, which included significant submarine mass wasting, debris, and turbidity flows. This event provided an excellent opportunity to investigate the effects of large-scale natural disturbance on benthic ecosystems. Benthic meiofauna community structure before and after the event was analysed from a time series of sediment cores collected 10 years and 6 years before, and 10 weeks, 10 months, and 4 years after the disturbance. Immediately after the 2016 event abundances of all meiofauna dramatically decreased. Four years later the meiofauna community had recovered and was no longer distinguishable from the pre-event community. However, the nematode component of the community was similar, but not fully comparable to the pre-event community by 4 years after the disturbance. Community recovery was systematically correlated to changes in the physical characteristics of the habitat caused by the disturbance, using physical and biochemical variables derived from sediment cores, namely: sediment texture, organic matter, and pigment content. While these environmental variables explained relatively little of the overall variability in meiofauna community structure, particle size, food availability and quality were significant components. The minimum threshold time for the meiofauna community to fully recover was estimated to be between 3.9 and 4.7 years, although the predicted recovery time for the nematode community was longer, between 4.6 and 5 years. We consider the management implications of this study in comparison to the few studies of large-scale disturbances in the deep sea, in terms of their relevance to the efficacy of the marine reserve that encompasses Kaikōura Canyon, along with potential implications for our understanding of the impacts of anthropogenic seafloor disturbances, such as seabed mining.

## INTRODUCTION

Disturbance is a key process that underpins the structure of all marine communities (*Sousa, 2001*). By creating heterogeneity and redistributing limiting resources (space, refuge, nutrients, *etc.*) disturbances structure ecological succession, increase habitat variability, and enhance biodiversity (*Sousa, 1984*; *Willig & Walker, 1999*). Many physical and biological factors determine the rate and pattern of resilience, resistance, and/or recovery of a community after a disturbance (*Sousa, 1984*). Here, as in *Bigham et al. (2023a)*; *Bigham et al. (2023b)*, resilience refers to the amount of disturbance that an ecosystem or its components can experience before changing to an alternative state, which is sometimes referred to as ecological resilience (*Holling, 1996*). Resistance is defined as the ability of an ecosystem, or its components, to remain unchanged from its initial state despite a disturbance (*Walker et al., 2004*). In contrast, recovery is defined as the return time after a disturbance for an ecosystem, or its components, to attain a stable state (*Folke et al., 2004*). Some of the largest benthic disturbances in the marine environment are caused by subaqueous sediment-density flows, which occur worldwide (*Bigham et al., 2021*). Sediment density flows occur when the material in submarine landslides mixes with water and creates high-density parcels of turbid water that travel downslope beneath less dense water (*Kuenen & Migliorini, 1950*; *Talling, 2014*). These turbulent, sediment-laden gravity flows are hydrodynamically complex, and a single event can contain multiple flow types with spatial and temporal variability (*Haughton, Barker & McCaffrey, 2003*; *Talling et al., 2007*; *Paull et al., 2018*). As such, many terms and classification schemes have been proposed to differentiate and recognise flow types, although confusion around the interpretation and application of these terms persists (*Kuenen & Migliorini, 1950*; *Lowe, 1979*; *Talling, Paull & Piper, 2013*). Herein, as in *Bigham et al. (2023a)*; *Bigham et al. (2023b)*, the term "turbidity flow" *sensu stricto Kuenen & Migliorini (1950)* will mainly be used because it is the commonly used overarching term for sediment density flows in the ecological literature (cf. *Bigham et al., 2021*).

Turbidity flows impact the benthic faunal communities in their path through both erosional and depositional processes (*Bigham et al., 2021*), but it is not clear to what extent these communities are resilient to the impacts of these different disturbances. Studies from the 2011 Tōhoku Earthquake showed rapid recovery (within 1.5 years) of the meiofaunal communities following a triggered turbidity flow (*Kitahashi et al., 2014*; *Kitahashi et al., 2016*; *Nomaki et al., 2016*). Turbidity flows pose a particular recolonisation challenge to meiofauna because they typically can only migrate laterally into relatively small, disturbed patches (*Chandler & Fleeger, 1983*; *Gallucci et al., 2008*; *Gollner, Miljutina & Bright, 2013*) and otherwise must be dispersed passively (*Ptatscheck & Traunspurger, 2020*). Despite these functional limitations and the potential for large-scale disturbances from turbidity flows, researchers of the Tōhoku studies hypothesised that meiofauna were more resilient to turbidity flows than macrofauna due to their faster turnover times and lower sensitivity to changes in environmental factors (*Kitahashi et al., 2014*; *Kitahashi et al., 2016*; *Nomaki et al., 2016*). Studies of turbidity flows hundreds to thousands of years old have suggested that the impact of the disturbance is still detectable thousands of years after the event

for all three benthic metazoan size classes (meio-, macro- and megafauna), although results were somewhat ambiguous for meiofauna specifically (*e.g.*, *Briggs, Richardson & Young, 1996*; *Griggs, Carey & Kulm, 1969*; *Woods & Tietjen, 1985*; *Lambshead et al., 2001*). Furthermore, such studies of older turbidity flows are often confounded by local patterns of surface productivity and terrigenous inputs that have occurred in the intervening time (*Richardson, Briggs & Young, 1985*; *Richardson & Young, 1987*; *Thurston et al., 1994*; *Briggs, Richardson & Young, 1996*; *Thurston, Rice & Bett, 1998*). Many studies, even ones of more recent turbidity flows, lack sufficient pre-disturbance data to fully interpret the impacts of turbidity flows on benthic communities (*Bigham et al., 2021*). Kaikōura Canyon off eastern New Zealand is the site of a recent and large earthquake-triggered turbidity flow (*Mountjoy et al., 2018*), with the additional context of pre-event benthic data (*De Leo et al., 2010*; *Leduc et al., 2014*).

Kaikōura Canyon on the northeastern side of the South Island, New Zealand, has been dubbed a benthic productivity hotspot due to an abundant macro- and megafaunal biological community with biomasses 100 times higher than those seen in (non-chemosynthetic) deep-sea habitats below 500 m (*De Leo et al., 2010*). The canyon also supports a distinct nematode community in response to high food availability and high frequency of disturbance, and which contributes significantly to regional meiofaunal diversity (*Leduc et al., 2014*). High organic carbon content and elevated meiofaunal biomass in Kaikōura Canyon, relative to another New Zealand canyon on the opposite side of the South Island (Hokitika Canyon), was inferred to be related to land-derived organic matter as a dominant food source (*Leduc et al., 2020*). On 14th November 2016, the Mw 7.8 Kaikōura Earthquake triggered a highly complex "full canyon-flushing event" that reshaped the canyon floor and transported an estimated 850 metric megatons of sediment and 7.5 metric megatons of organic carbon through the canyon, and into the slope basin *via* the Hikurangi Channel (*Mountjoy et al., 2018*). The flushing event was geomorphologically complex, including submarine landslides and other local mass wasting episodes that generated debris and cascading turbidity flows down the canyon walls, forming a large flow down the canyon axis (for simplicity, as above, this event is hereafter referred to as the "turbidity flow"). Analysis of time-series imagery from the canyon found that the seafloor and near-seafloor megafauna community structure was recovering, with full recovery predicted 4.6–5.2 years after the turbidity flow (*Bigham et al., 2023a*). Analysis of the macrofauna community structure in the canyon substrate also suggested ongoing recovery with full recovery predicted 5.6–6.7 years after the turbidity flow (*Bigham et al., 2023b*). With comprehensive repeat datasets from before and after the turbidity flow, Kaikōura Canyon provides a unique opportunity to also explore meiofauna community resilience to the impact of turbidity flows on deep-sea fauna.

The present study compares meiofauna community structure in Kaikōura Canyon before and after the turbidity flow event in 2016 to determine the community response to the event in relation to changes in the environmental characteristics of the habitat caused by the disturbance. The management implications for the Hikurangi Marine Reserve, which envelopes the Kaikōura Canyon head and much of its middle reach, and the potential for turbidity flows to be considered as proxies for predicting the impacts of widespread physical

disturbances caused by large-scale deep-sea mining in the future are also considered, as they were for the megafauna and macrofauna studies (*Muñoz Royo et al., 2022*; *Bigham et al., 2023a*; *Bigham et al., 2023b*).

## MATERIALS & METHODS

### Site descriptions

Kaikōura Canyon is located off the northeastern coast of New Zealand's South Island (Fig. 1). The canyon is 60 km long, broadly sinuous, ranges in depth from 20 m to >2,000 m, is generally U-shaped in profile, and is the primary headwater source for the 1,500-km long Hikurangi Channel, which transports sediments from the axial mountain chain of the South Island to a distal abyssal fan-drift northeast of New Zealand (*Lewis, 1994*). The Kaikōura Canyon incises into the narrow continental shelf, the head shoaling to within 500 m of the shore (*Lewis & Barnes, 1999*). The November 2016 "full canyon flushing" event caused significant erosion and deposition on the canyon floor (*Mountjoy et al., 2018*). These areas of impact, indicated by the measured bathymetric changes (Fig. 1), informed the post-event benthic sampling campaigns, as did the location of pre-disturbance sampling in the canyon. Impacts to the megafauna and macrofauna communities from this same site and disturbance can be found in *Bigham et al. (2023a)*; *Bigham et al. (2023b)*.

### Sampling and sample processing

Field sampling was undertaken under the General Special Permit (841 and 842) issued by Fisheries New Zealand to the National Institute of Water and Atmospheric Research. Sediment core samples were collected from the R/V *Tangaroa* using an Ocean Instruments MC-800A multicorer (internal core diameter = 9.52 cm), which can collect up to eight cores in a single deployment. These cores were processed for different analyses, such as meiofauna community or sediment characteristics as well as macrofauna community (see *Bigham et al., 2023b*). Samples from eight sites along the axis of Kaikōura Canyon (depth range 400–1,300 m) were collected 6 years before the turbidity flow and 10 months and 4 years after the event (voyages TAN1006, TAN1708, and TAN2011, respectively). Samples from two of the eight main sites were also collected 10 years before and 10 weeks after the disturbance (TAN0616 and TAN1701, respectively) (Fig. 1B). See Table 1 for site and sampling details and Fig. 1B for site locations. Bathymetric difference mapping by *Mountjoy et al. (2018)* identified zones along the canyon length after the flushing event that were net erosional (downcut) or depositional (elevation gain). As can be seen on the map in Fig. 1, the samples come from sites broadly occupying two different disturbance regimes—most of the samples come from sites where the net change was erosional, but two of the deepest sample sites, K06 and K07 are from depositional zones.

For this study, one core per site was analysed for meiofaunal community analyses, except for samples from 10 weeks after (TAN1701) where two cores were analysed and 6 years before (TAN1006) where two to three cores were analysed (see Table 1 for details). Each meiofauna sample consists of one syringe subcore (internal diameter 26 or 29 mm) to five cm depth. The subcore was sectioned at 0–1 cm and 1–5 cm depth and preserved in 100% buffered formalin, though only depth integrated 0–5 cm results are reported here.
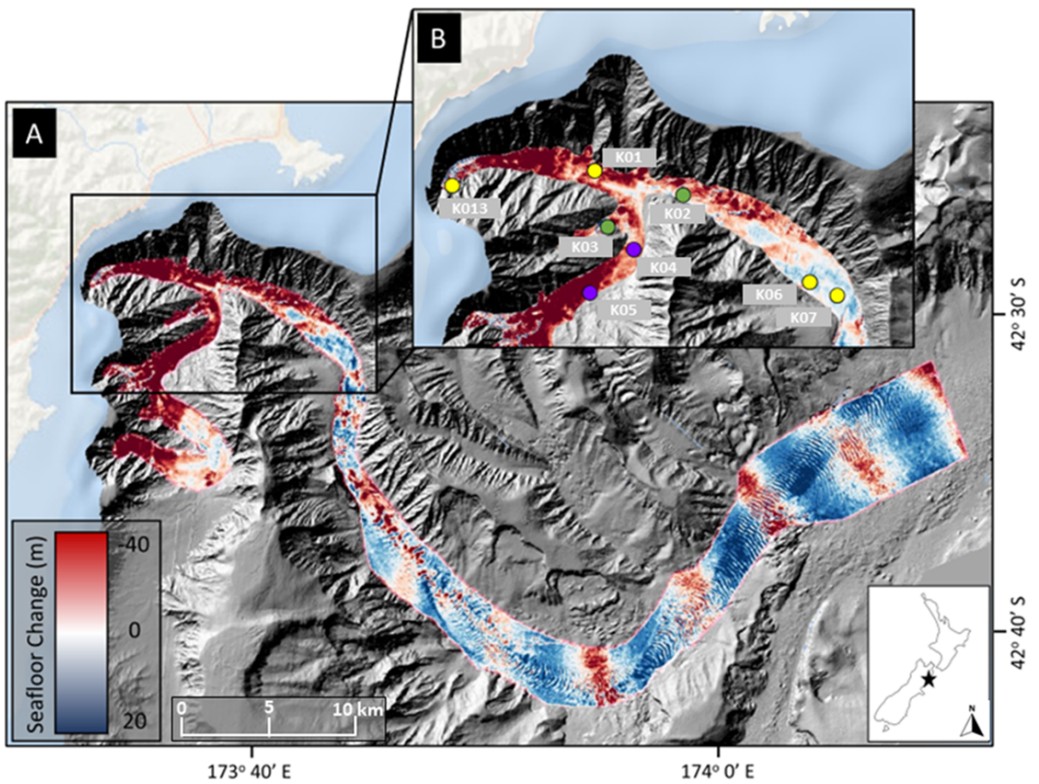

**Figure 1** **Map of sampling locations.** Location of sampling sites in Kaikōura Canyon overlayed on canyon flushing-induced bathymetric changes. (A) Magnitude of erosion and deposition (seafloor change) within Kaikōura Canyon caused by the canyon flushing triggered by the Kaikōura Earthquake, measured by the differencing the pre- and post-earthquake bathymetry data sets (*Mountjoy et al., 2018*). (B) Location of the time-series of multicorer sampling sites (yellow circles = sampled in 2010, late 2017, and 2020; purple circles = sampled in 2006 in addition to other time points; green circles = sampled in early 2017 in addition to other time points) within the head of Kaikōura Canyon. Inset shows the location of Kaikōura Canyon (star) relative to New Zealand. Some of the red (erosional) banding evident along the bottom reach of Kaikōura Canyon is an artefact of higher levels of uncertainty in bathymetric differencing for overlapping multibeam coverages (for more detail see *Mountjoy et al., 2018*). Image source credit: *Bigham et al. (2023b)* CC-BY 4.0.

In the laboratory, samples were sieved through a one mm mesh with fresh water to remove macrofauna, and through a 45 μm mesh size to retain meiofauna. Ludox flotation was used to extract meiofauna from the remaining sediment (*Somerfield & Warwick, 1996*). Samples were transferred to a Bogorov tray and all meiofauna present in the sample were identified to major taxa (*e.g.*, nematodes, annelids, harpacticoid copepods, kinorhynchs) and counted using a compound microscope (x100 magnification).

All nematodes from the samples were transferred to a mixture of dilute ethanol and glycerol in a cavity block and left under a fume hood for at least 48 h to allow water and ethanol to evaporate, leaving the sample material in pure glycerol (*Somerfield & Warwick, 1996*). Nematodes were mounted on slides in pure glycerol and sealed with paraffin wax. Nematode body volumes were estimated using ImageJ to measure length and maximum body width for all eight sites. Body volumes were converted to dry weight (DW) based

Bigham et al. (2024), *PeerJ*, DOI 10.7717/peerj.17367

**Table 1 Multicore sampling site details, including depth ranges and number of cores used for meiofauna and sediment analyses.**

| Time point | Date | Voyage | Station number | Site | Depth (m) | Latitude | Longitude | Number of cores for meiofauna | Number of cores for sediment | Reference |
|---|---|---|---|---|---|---|---|---|---|---|
| 10 years before | November 2006 | TAN0616 | 98 | K04 | 1061 | −42.512 | 173.633 | 2 | – | This study |
| | | | 105 | K05 | 1,020 | −42.523 | 173.621 | 2 | – | This study |
| 6 years before | May 2010 | TAN1006 | 6 | K13 | 404 | −42.490 | 173.551 | 3 | 1 | *Leduc et al. (2020)* |
| | | | 4 | K01 | 1,017 | −42.484 | 173.615 | 2 | 1 | *Leduc et al. (2020)* |
| | | | 3 | K02 | 989 | −42.524 | 173.613 | 2 | 1 | *Leduc et al. (2020)* |
| | | | 14 | K03 | 1,032 | −42.504 | 173.619 | 2 | 1 | *Leduc et al. (2020)* |
| | | | 7 | K04 | 1,061 | −42.508 | 173.633 | 2 | 1 | *Leduc et al. (2020)* |
| | | | 8 | K05 | 1,127 | −42.492 | 173.657 | 2 | 1 | *Leduc et al. (2020)* |
| | | | 2 | K06 | 1,289 | −42.520 | 173.712 | 3 | 1 | *Leduc et al. (2020)* |
| | | | 11 | K07 | 1,320 | −42.524 | 173.736 | 2 | 1 | *Leduc et al. (2020)* |
| 10 weeks after | February 2017 | TAN1701 | 181 | K02 | 1,186 | −42.492 | 173.653 | 2 | 1 | This study |
| | | | 182 | K03 | 1,036 | −42.501 | 173.625 | 2 | 1 | This study |
| 10 months after | September 2017 | TAN1708 | 130, 131 | K13 | 422 | −42.490 | 173.551 | 1 | 1 | This study |
| | | | 127 | K01 | 994 | −42.485 | 173.615 | 1 | 1 | This study |
| | | | 6 | K02 | 1,188 | −42.492 | 173.653 | 1 | 1 | This study |
| | | | 12, 11 | K03 | 1,000 | −42.502 | 173.622 | 1 | 1 | This study |
| | | | 16 | K04 | 1,069 | −42.510 | 173.632 | 1 | 1 | This study |
| | | | 28 | K05 | 1014 | −42.524 | 173.613 | 1 | 1 | This study |
| | | | 75 | K06 | 1,230 | −42.520 | 173.712 | 1 | 1 | This study |
| | | | 70 | K07 | 1,298 | −42.525 | 173.725 | 1 | 1 | This study |
| 4 years after | October 2020 | TAN2011 | 79 | K13 | 425 | −42.490 | 173.551 | 1 | 1 | This study |
| | | | 58 | K01 | 1,048 | −42.485 | 173.615 | 1 | 1 | This study |
| | | | 38 | K02 | 1,190 | −42.492 | 173.653 | 1 | 1 | This study |
| | | | 35 | K03 | 1,049 | −42.502 | 173.622 | 1 | 1 | This study |
| | | | 47 | K04 | 1,068 | −42.510 | 173.632 | 1 | 1 | This study |
| | | | 50 | K05 | 1,015 | −42.524 | 173.613 | 1 | 1 | This study |
| | | | 86 | K06 | 1,293 | −42.520 | 173.712 | 1 | 1 | This study |
| | | | 83 | K07 | 1,312 | −42.525 | 173.725 | 1 | 1 | This study |
on a relative density of 1.13 and a DW: wet weight (WW) ratio of 0.25 (*Feller & Warwick, 1988*). Nematodes present in the samples from sites K2 and K3 (green circles in Fig. 1B) 6 years before and 10 weeks, 10 months, and 4 years after the turbidity flow were identified to species/morphospecies using a compound microscope (x1,000 magnification) and percentage of juveniles to adults was recorded. The percentage of juveniles was averaged for each time point.

One additional core per site and per time point was analysed for sediment parameters. These are the same environmental parameters used in *Bigham et al. (2023b)* for comparison with the macrofaunal community in the canyon and full details can be found there.

## Statistical analysis

Many of the same statistical analyses used in this study were also used in *Bigham et al. (2023a)*, *Bigham et al. (2023b)* and further details on the methodology can be found there. As was the case in the previous two studies, unless otherwise noted statistical analyses were carried out using routines in PRIMER 7 (*Clarke & Gorley, 2018*) with PERMANOVA + (*Anderson, Gorley & Clarke, 2008*).

Meiofaunal communities typically comprise both permanent meiofaunal taxa (*e.g.*, nematodes and harpacticoid copepods, which spend their entire life cycle in the sediment) and temporary meiofaunal taxa (*i.e.,* juveniles of macrofaunal-sized taxa, such as most polychaetes and molluscs, which typically have a pelagic larval stage) (*Warwick, 1988*). Temporary meiofauna are typically excluded from analyses of meiofauna communities due to being larger, having a highly patchy distribution, and occurring at low densities relative to permanent meiofauna (*Higgins & Thiel, 1988*). However, juvenile macrofauna are an important indicator of overall community response to disturbance from the turbidity flow as they provide information on macrofauna taxa that would not be captured in the macrofauna analyses alone (*Bigham et al., 2023b*). Since this study is concerned with the full community response, both permanent and temporary meiofauna were used in the community analysis.

Nematodes are typically the most abundant taxon of meiofauna communities (*Giere, 2008*). They are an important group for indicating impact and recovery from disturbance and specific genera are used as indicator taxa for disturbance (*Boyd, Rees & Richardson, 2000*; *Lambshead et al., 2001*; *van Gaever et al., 2009*; *Leduc & Pilditch, 2013*; *Semprucci, Losi & Moreno, 2015*; *Zeppilli et al., 2015*; *Ingels et al., 2020*). However, due to the time constraints of high-level taxonomic identification (*Miljutin et al., 2010*) only the two sites (K2 and K3) with the most time points had nematodes identified to species level. Therefore, where appropriate, statistical analyses were undertaken on both the complete meiofauna community and nematode species community separately.

It was preferred to use a single time point to represent 'before' the turbidity flow event for the statistical analysis. Therefore, an analysis of similarity test was used to confirm there was no significant difference in community structure between data from 10 years and 6 years before (ANOSIM, $R = 0.466$, $p = 0.089$, Number of permutations: 45). The exception to using the combined Before data was for the Distance-based linear models (DISTLM)

only data from 6 years before was used because environmental data was not available from 10 years before the disturbance event.

## Community structure

Analyses were run on data from the 0–5 cm sediment depth layer for both the non-species level meiofauna data, including nematodes (referred to hereafter as "meiofauna" community data), as well as the nematodes identified to species level (referred to hereafter as "nematode" community data). For the meiofauna community analyses replicate cores from the same site at the same time point were averaged (See Table 1 for details). For the nematode community there were two cores from both sites 10 weeks after these were kept separate for the analyses to.

Similarity matrices for the multivariate community structure data were made using the zero-adjusted Bray–Curtis similarity measure of square root transformed abundances (*Clarke & Gorley, 2018*). Variability in the community structure through times was tested using PERMANOVAs with Type III (partial) sums of squares, unrestricted permutation of raw data and 9,999 permutations. A pair-wise PERMANOVA was only run on the meiofauna data, since there were only two sites per time point for the nematode data. The results of these multivariate community structure analysis were visualized using two-dimensional non-metric multi-dimensional (nMDS) plots. The centroids, the point at the centre of the data cloud, provide a simplified view of the overall patterns. The SIMPER routine was run on meiofauna and nematode data to determine the contribution of taxa to within and between community similarity for each time point. A cut-off of 70% was used in the SIMPER routine to identify key taxa contributing to the similarity/dissimilarity. Further, the RELATE test of cyclicity (correlation method: Spearman rank ($\rho$), Number of permutations: 9,999), MVDISP, and PERMDISP (number of permutations: 9,999) routines were used to evaluate the patterns observed in the meiofauna nMDS plot. Additionally, PERMDISP (number of permutations: 9,999) was used to determine the significance of differences in the multivariate dispersion (*Anderson, Gorley & Clarke, 2008*). It was not possible to run the RELATE test, MVDISP, or PERMDISP on the nematode data due to the small sample size.

## Environmental drivers

Distance-based linear models (DISTLM) were used to assess the effect of environmental parameters on the meiofauna and nematode community structure. The predictor variables were those sediment parameters detailed in *Bigham et al. (2023b)*, which were used to characterise food availability and physical sediment habitat, as well as the water depth at which the sample was taken. Correlation between environmental variables was checked before running the DISTLM using draftsman plots and correlation matrices. When Pearson's r was > 0.8 between variables, one of the variables was excluded from the analysis; if more than one variable correlated with others, the variable with the most correlations was kept. Both DISTLMs were run with stepwise variable selection, the Akaike Information Criterion (AIC) and 9,999 permutations. Both marginal and sequential tests were conducted. Marginal tests examine a single variable separately, while the sequential test

takes in to account the previously tested variables when examining each variable (*Anderson, Gorley & Clarke, 2008*). The best models proposed by the DISTLM were visualized with distance-based redundancy analysis (dbRDA) ordination plots.

## Predicting recovery

While a result of no significance difference between the meiofauna community structure before and 4 years after the turbidity flow indicates that recovery has occurred (see Results) it does not provide an estimate of the trajectory of the recovery, nor an indication when recovery may have occurred before the time point that the non-significant result was apparent. To estimate the recovery trajectory, rates of recovery were predicted by fitting three common models of population growth (linear, exponential, logistic; *Lundquist et al., 2010*; *Solé et al., 2010*) using the generalized linear model and nonlinear least squares functions in R (*R Core Team, 2022*). The models were fitted to the observed similarity at the three post-event time points for the 0–5 cm sediment layer for the meiofauna and the nematode data. The meiofauna and nematode community were predicted to be recovered when they at least reached the level of within-group similarity exhibited by the pre-turbidity flow community (*i.e.,* 79% and 46.1%, respectively).

# RESULTS

## Community structure

The Kaikōura Canyon meiofauna community has recovered overall following the disturbance by the turbidity flow. The community structure differed significantly between prior to the disturbance and 10 months after the event (PERMANOVA, $p < 0.001$, Table 2), but by 4 years after the turbidity flow the community structure was no longer significantly different ($p = 0.1703$, Table 2). In contrast, the main PERMANOVA test for the nematode community indicates a significant difference in the community structure among all time points, for the two-site subset of data ($df = 3$, SS = 13,174, MS = 4,391.2, Pseudo-F = 2.4428, P(perm) = 0.0014). The recovery pattern of the meiofauna and nematode communities are illustrated in the multivariate ordinations of community similarity. The nMDS plots show clustering of samples by time point, with samples from Before the turbidity flow and 4 years after clustered most distinctly on the left-hand side of the plot, and samples taken 10 weeks and 10 months after the event spread out on the right-hand side (Figs. 2A and 2C). The centroids of the meiofauna and nematode community sample data, with trajectories overlaid, are also displayed to provide a simplified illustration of the pattern (Figs. 2B and 2D). The test for cyclicity for the meiofauna community was not significant (rho = 0.061, $p = 7.42\%$), meaning that the meiofauna community's pattern of recovery was not comparable to a simple, equal distance cyclical recovery. Community variability (dispersion) for the meiofauna community was greatest in the weeks and months after the turbidity flow (10 weeks: 1.165, 10 months after: 1.56), then decreased as recovery progressed towards the original community structure (4 years after: 0.72) and was significantly different between all time points (Pseudo-$F$ = 7.1157, P(perm) < 0.001).

The SIMPER analysis of the meiofauna community found that nematodes consistently contributed between 66 and 77% of the within community similarity, with copepods also

**Table 2  Results of the main and pair-wise PERMANOVA tests for differences between time points for meiofauna community structure.**

|  |  | Pseudo-F/t | P(perm) | Permutations |
|---|---|---|---|---|
| Main |  | 10.545 | 0.0001 | 9,950 |
| Pair-wise | Before, 10 weeks after | 4.428 | 0.0176 | 66 |
|  | Before, 10 months after | 3.991 | 0.0002 | 8,875 |
|  | Before, 4 years after | 1.293 | 0.1703 | 8,874 |
|  | 10 weeks after, 10 months after | 0.751 | 0.6424 | 45 |
|  | 10 weeks after, 4 years after | 4.719 | 0.0207 | 45 |
|  | 10 months after, 4 years after | 3.488 | 0.0005 | 5,086 |

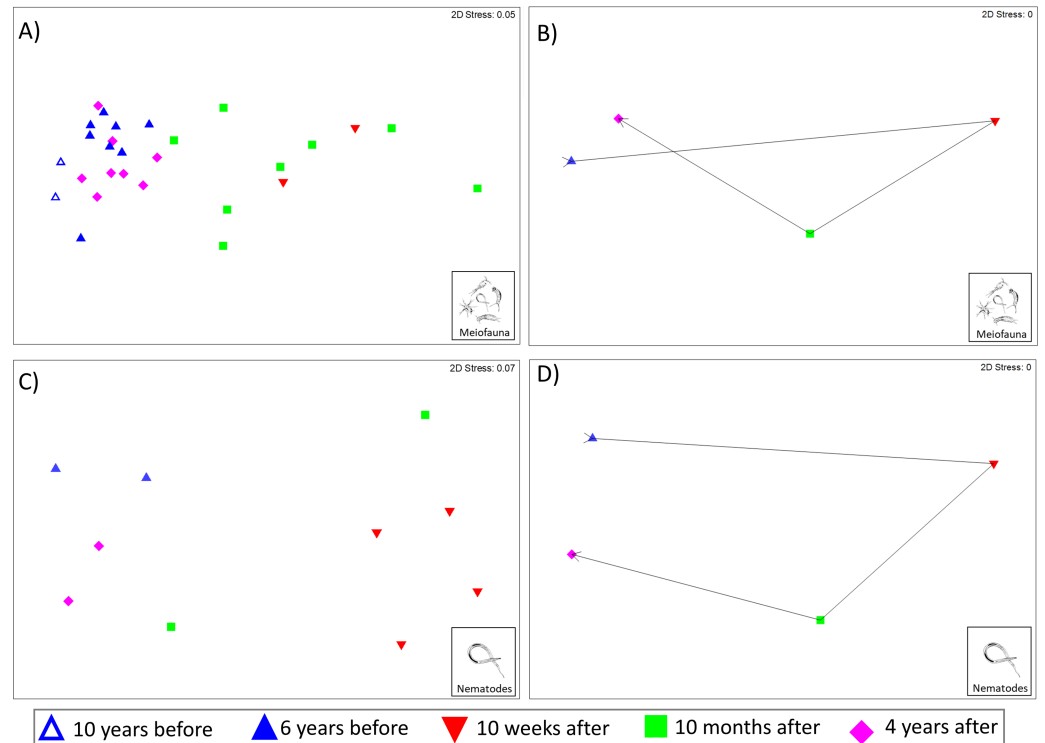

**Figure 2  Non-metric multidimensional scaling (nMDS) plots of meiofauna and nematode community structure.** Non-metric multidimensional scaling (nMDS) plots of community structure: (A) meiofauna, (B) meiofauna centroids, (C) nematodes, and (D) nematode centroids before the turbidity flow and at 10 weeks, 10 months, and 4 years after the disturbance in Kaikōura Canyon. For meiofauna centroids (B) data from 10 and 6 years before has been combined into a single "Before" centroid. Similarities were calculated from zero adjusted, square root transformed fauna abundances for both community levels. All stress values are below 0.2, indicating that the plots are acceptable representations of the similarity patterns.

being key contributors 10 weeks after the turbidity flow and nauplii key contributors 4 years after the disturbance event (Table 3). The nematode community SIMPER analysis found that between four and nine species of nematodes explained within time point

**Table 3** **SIMPER analysis results for the meiofauna community for each time point.** SIMPER analysis results for the meiofauna community indicating average within time point community similarity and the contribution of individual taxa contributing 70% or more to within time point community similarity. Avg. Abundance is the average abundance of individuals standardized to 10 cm² of seafloor area.

| Time point | Average similarity | Taxon | % Contribution | Avg. Abundance |
|---|---|---|---|---|
| Before (10 and 6 years) | 79.44 | Nematodes | 71.94 | 2,285.80 |
| 10 weeks after | 74.46 | Nematodes | 66.08 | 156.75 |
| | | Copepods | 15.89 | 15.21 |
| 10 months after | 59.28 | Nematodes | 76.69 | 367.49 |
| 4 years after | 81.57 | Nematodes | 69.32 | 2,027.70 |
| | | Nauplii | 8.98 | 55.20 |

community similarity. *Hopperia beaglense* had the highest contribution to within time point community similarity Before the turbidity flow, while *Daptonema* sp. 18 was the highest contributor 10 weeks, 10 months, and 4 years after the event (Table 4).

At the meiofauna community level, community dissimilarity among time points was explained by up to five taxa. Dissimilarity was highest between samples from Before and 10 weeks after the turbidity flow at 57.75%, with only a small decrease in dissimilarity to 51.67% between Before and 10 months after the event. Ten weeks and 10 months after the disturbance the average abundances of all key contributing taxa were lower than the average abundances from before the event. Nematodes and kinorhynchs were key contributors to the differences between the community at 10 weeks after and Before the turbidity flow. Along with nematodes and kinorhynchs, nauplii contributed to differences between the Before community and the community 10 months after the event. Dissimilarity was lowest, at 20.75%, between samples from before the turbidity flow and 4 years after the event. Nauplii, copepods, and copepods were key contributors to dissimilarity between the community 4 years after the turbidity flow and the Before community. Four years after the event, nematode average abundances were similar to their pre-disturbance levels, while copepod and nauplii average abundances exceeded pre-event levels. Kinorhynch and gastrotrich abundances at 4 years after the event remained depressed compared to before the turbidity flow (Table 5).

Nematode community dissimilarity among time points can be explained by many species. *Sabatieria* sp. A, *Hopperia beaglense*, *Microlaimus* sp. 34, *Cervonema kaikouraensis*, and *Daptonema* sp. 18 were consistently among the highest contributors to the observed dissimilarity. Dissimilarity was highest, at 81.79%, between the community Before the disturbance and the community sampled 10 weeks after the turbidity flow. This difference in the two communities was represented by a large decrease in the abundance of all key contributory taxa. Dissimilarity decreased to 77.50% between the Before community and the community 10 months after the turbidity flow. At this stage in community recovery the dissimilarity was characterized with continued low average abundances for most key contributory taxa except for *Daptonema* sp. 18, which had nearly tripled in abundance from its pre-event average abundance. *Campylaimus* sp. 6, *Leptolaimus* sp. 14, and *Sphaerolaimus* sp. 1 were not observed 10 months after the disturbance despite being present 10 weeks

**Table 4  SIMPER analysis results for the nematode community for each time point.** SIMPER analysis results for the nematode community indicating average within time point community similarity and the contribution of individual species contributing 70% or more to within time point community similarity. Avg. Abundance is the average abundance of individuals standardized to 10 cm² of seafloor area.

| Time point | Average similarity | Species | % contribution | Avg. abundance |
|---|---|---|---|---|
| Before (6 years) | 46.11 | *Hopperia beaglense* | 12.99 | 178.22 |
| | | *Cervonema kaikouraensis* | 10.74 | 126.56 |
| | | *Campylaimus* sp. 6 | 8.32 | 96.83 |
| | | *Leptolaimus* sp. 14 | 8.32 | 78.50 |
| | | *Sabatieria* sp. A | 8.22 | 354.57 |
| | | *Daptonema* sp. 18 | 7.12 | 83.72 |
| | | *Metalinhomoeus* sp. 1 | 7.12 | 50.84 |
| | | *Sabatieria* sp. 12 | 7.12 | 98.80 |
| | | *Sphaerolaimus* sp. 1 | 7.12 | 59.60 |
| 10 weeks after | 42.77 | *Daptonema* sp. 18 | 19.26 | 20.43 |
| | | *Sabatieria* sp. A | 15.79 | 11.22 |
| | | *Metacyatholaimus* sp. 1 | 7.81 | 4.08 |
| | | *Cervonema kaikouraensis* | 7.12 | 1.85 |
| | | *Monhysteridae* sp. 35 | 7.12 | 1.51 |
| | | *Vasostoma hexodontium* | 7.12 | 1.51 |
| | | *Daptonema* sp. 23 | 3.87 | 4.45 |
| | | *Paramonohystera* sp. 1 | 3.87 | 2.72 |
| 10 months after | 22.35 | *Daptonema* sp. 18 | 27.25 | 242.42 |
| | | *Sabatieria* sp. 12 | 23.03 | 8.41 |
| | | *Daptonema* sp. 21 | 14.56 | 18.23 |
| | | *Sabatieria* sp. A | 14.56 | 18.23 |
| 4 years after | 47.77 | *Daptonema* sp. 18 | 30.52 | 1,012.51 |
| | | *Sabatieria* sp. A | 13.82 | 273.57 |
| | | *Microlaimus* sp. 34 | 10.27 | 70.90 |
| | | *Cervonema kaikouraensis* | 8.38 | 51.98 |
| | | *Daptonema* sp. 21 | 8.38 | 44.76 |

after the event. Community dissimilarity was lowest, at 59.71%, between samples from Before the turbidity flow and 4 years after the event. By 4 years after the turbidity flow, the average densities of contributory species had begun to increase, though had not attained pre-event levels. In contrast, *Leptolaimus* sp. 14 continued to be absent, while *Daptonema* sp. 18 was now 12 times pre-event levels, and *Endeolophos* sp. 3 had increased to an average abundance three times pre-event levels (Table 6).

## Environmental drivers

Of the eight environmental variables included in the DISTLM analysis the marginal test identified two variables (TOM% and PN% (borderline $p$-value; 0.049)) as significant ($p$-value $\leq 0.05$) explanatory variables for the meiofauna community structure, and one variable (C:N (molar)) as significant for the nematode community structure. The best DISTLM model (AIC = 152.82, $R^2 = 0.28971$, RSS = 12,481) for meiofauna community

**Table 5  SIMPER analysis results for the meiofauna community between time points.** SIMPER analysis results for the meiofauna community indicating average among time point community dissimilarity and the contribution of individual taxa contributing 70% of more to among time point community dissimilarity. Avg. Abundance is the average abundance of individuals standardized to 10 cm² of seafloor area.

| Time points | Average dissimilarity | Taxon | % contribution | Time 1 avg. abundance | Time 2 avg. abundance |
|---|---|---|---|---|---|
| Before (10 and 6 years), 10 weeks after | 57.75 | Nematodes | 62.57 | 2,285.80 | 156.75 |
| | | Kinorhynchs | 1.76 | 43.69 | 2.46 |
| Before (10 and 6 years), 10 months after | 51.67 | Nematodes | 55.20 | 2,285.80 | 367.49 |
| | | Kinorhynchs | 10.75 | 43.69 | 0.17 |
| | | Nauplii | 8.09 | 36.36 | 3.80 |
| Before (10 and 6 years), 4 years after | 20.75 | Nematodes | 24.56 | 2,285.80 | 2,027.70 |
| | | Nauplii | 15.27 | 36.36 | 55.20 |
| | | Copepods | 13.36 | 37.09 | 55.06 |
| | | Kinorhynchs | 12.69 | 43.69 | 7.95 |
| | | Gastrotrichs | 6.81 | 2.86 | 1.90 |
| 10 weeks after, 10 months after | 33.55 | Nematodes | 49.75 | 156.75 | 367.49 |
| | | Copepods | 17.10 | 15.21 | 12.60 |
| | | Nauplii | 11.36 | 2.13 | 3.80 |
| 10 weeks after, 4 years after | 54.98 | Nematodes | 62.88 | 156.75 | 2,027.70 |
| | | Nauplii | 11.58 | 2.13 | 55.20 |
| 10 months after, 4 years after | 47.95 | Nematodes | 55.83 | 367.49 | 2,027.70 |
| | | Nauplii | 11.88 | 3.80 | 55.20 |
| | | Copepods | 10.14 | 12.60 | 55.06 |

structure included 3 variables, only one (%TOM) of which was significantly correlated to the community structure and explained 14% of the sample variation across all time points (see sequential test under meiofauna in Table 7). While the best DISTLM model (AIC = 56.28, $R^2$ = 0.9117, RSS = 1,578.9) for the nematode community structure included 6 variables, of which only one (C:N (molar); explaining 33% of variation) was significantly correlated to the community structure and explained 91% of the sample variation across all time points (see sequential test under nematodes in Table 7).

The first two axes of the dbRDA plots explained 24.9% and 3.8% of total community variation for meiofauna, and 37.8% and 17.2% for nematodes (Fig. 3). For the meiofauna, dbRDA1 accounted for most of the variation among the samples; it was primarily correlated with %TOM, sediment Chl *a* concentrations, and C:N (molar). dbRDA2 accounts for a much smaller portion of the variation, primarily that for community variation between 6 years before the turbidity flow and the other time points and is correlated with negative sediment skewness and higher percentages of sediment less than 16 μm (Fig. 3A). For the nematodes, dbRDA1 accounts for the variation in samples between 10 weeks after

**Table 6  SIMPER analysis results for the nematode community between time points.** SIMPER analysis results for the nematode community indicating average among time point community dissimilarity and the contribution of individual species contributing 70% of more to among time point community dissimilarity. Avg. Abundance is the average abundance of individuals standardized to 10 cm² of seafloor area.

| Time points | Average dissimilarity | Species | % contribution | Time 1 avg. abundance | Time 2 avg. abundance |
|---|---|---|---|---|---|
| Before (6 years), 10 weeks after | 81.79 | *Sabatieria* sp. A | 7.61 | 354.57 | 11.22 |
| | | *Hopperia beaglense* | 6.51 | 178.22 | 0.19 |
| | | *Microlaimus* sp. 34 | 6.05 | 133.17 | 1.64 |
| | | *Cervonema kaikouraensis* | 5.04 | 126.56 | 1.85 |
| | | *Campylaimus* sp. 6 | 4.87 | 96.83 | 0.10 |
| | | *Sabatieria* sp. 12 | 4.58 | 98.80 | 0.55 |
| | | *Leptolaimus* sp. 14 | 4.37 | 78.50 | 0.10 |
| | | *Sphaerolaimus* sp. 1 | 3.59 | 59.60 | 0.38 |
| | | *Retrotheristus* sp. 5 | 3.35 | 58.37 | 0.94 |
| | | *Metalinhomoeus* sp. 1 | 3.13 | 50.84 | 0.94 |
| Before (6 years), 10 months after | 77.50 | *Sabatieria* sp. A | 7.38 | 354.57 | 18.23 |
| | | *Microlaimus* sp. 34 | 6.12 | 133.17 | 1.14 |
| | | *Hopperia beaglense* | 6.12 | 178.22 | 2.31 |
| | | *Daptonema* sp. 18 | 6.01 | 83.72 | 242.42 |
| | | *Campylaimus* sp. 6 | 5.09 | 96.83 | 0.00 |
| | | *Cervonema kaikouraensis* | 5.09 | 126.56 | 2.31 |
| | | *Leptolaimus* sp. 14 | 4.57 | 78.50 | 0.00 |
| | | *Sphaerolaimus* sp. 1 | 3.94 | 59.60 | 0.00 |
| | | *Sabatieria* sp. 12 | 3.56 | 98.80 | 8.41 |
| | | *Metalinhomoeus* sp. 1 | 3.16 | 50.84 | 0.76 |
| Before (6 years), 4 years after | 59.71 | *Daptonema* sp. 18 | 10.51 | 83.72 | 1,012.51 |
| | | *Microlaimus* sp. 34 | 5.32 | 133.17 | 70.90 |
| | | *Sabatieria* sp. A | 5.12 | 354.57 | 273.57 |
| | | *Leptolaimus* sp. 14 | 4.07 | 78.50 | 0.00 |
| | | *Campylaimus* sp. 6 | 3.31 | 96.83 | 7.56 |
| | | *Endeolophos* sp. 3 | 3.03 | 32.38 | 101.20 |
| 10 weeks after, 10 months after | 71.23 | *Daptonema* sp. 18 | 15.21 | 20.43 | 242.42 |
| | | *Daptonema* sp. 21 | 5.31 | 0.19 | 18.23 |
| | | *Endeolophos* sp. 3 | 4.03 | 0.10 | 10.56 |
| | | *Sabatieria* sp. A | 3.74 | 11.22 | 18.23 |
| | | *Daptonema* sp. 27 | 3.71 | 0.10 | 9.24 |
| | | *Sabatieria* sp. 12 | 3.31 | 0.55 | 8.41 |
| | | *Paramesonchium* sp. 2 | 3.06 | 0.00 | 4.58 |
| | | *Metacyatholaimus* sp. 1 | 3.04 | 4.08 | 0.00 |

**Table 6** (*continued*)

| Time points | Average dissimilarity | Species | % contribution | Time 1 avg. abundance | Time 2 avg. abundance |
|---|---|---|---|---|---|
| 10 weeks after, 4 years after | 82.55 | *Daptonema* sp. 18 | 16.30 | 20.43 | 1,012.51 |
| | | *Sabatieria* sp. A | 7.88 | 11.22 | 273.57 |
| | | *Endeolophos* sp. 3 | 5.62 | 0.10 | 101.20 |
| | | *Hopperia beaglense* | 4.63 | 0.19 | 71.74 |
| | | *Microlaimus* sp. 34 | 4.19 | 1.64 | 70.90 |
| | | *Daptonema* sp. 21 | 3.69 | 0.19 | 44.76 |
| | | *Cervonema kaikouraensis* | 3.42 | 1.85 | 51.98 |
| | | *Dichromadora* sp. 7 | 3.37 | 0.76 | 44.89 |
| | | Comesomatidae sp. 6 | 3.25 | 0.00 | 27.77 |
| | | *Sabatieria* sp. 12 | 3.22 | 0.55 | 27.77 |
| | | *Chromadora* sp. 1 | 3.13 | 0.10 | 32.72 |
| 10 months after, 4 years after | 69.10 | *Daptonema* sp. 18 | 13.03 | 242.42 | 1,012.51 |
| | | *Sabatieria* sp. A | 8.56 | 18.23 | 273.57 |
| | | *Microlaimus* sp. 34 | 4.83 | 1.14 | 70.90 |
| | | *Hopperia beaglense* | 4.68 | 2.31 | 71.74 |
| | | *Endeolophos* sp. 3 | 4.57 | 10.56 | 101.20 |
| | | *Dichromadora* sp. 7 | 3.95 | 0.38 | 44.89 |
| | | *Cervonema kaikouraensis* | 3.89 | 2.31 | 51.98 |
| | | *Chromadora* sp. 1 | 3.78 | 0.00 | 32.72 |
| | | Comesomatidae sp. 6 | 3.70 | 0.00 | 27.77 |
| | | *Sabatieria* sp. 12 | 3.60 | 8.41 | 27.77 |

the turbidity flow and the other three time points; it is primarily correlated with C:N (molar) and the ratio of Chl *a* to phaeopigments. dbRDA2 also accounts for a large amount of variation, primarily between the samples taken 6 years before and 10 months and 4 years after the disturbance event; this axis correlates to Chl *a* concentrations and percent particulate nitrogen (Fig. 3B). The sediment samples taken 10 weeks, 10 months after, and 4 years after the turbidity flow all had higher percentages of TOM, nitrogen, and sediment particles greater than 16 $\mu$m compared to 6 years before the event. In contrast, all had lower concentrations of Chl *a*, ratio of Chl *a* to phaeopigments, ratio of C:N (molar), and a slightly negative skewed distribution of sediment grain size (Figs. 4A–4G).

Additionally, highly correlated but removed variables would likely also explain the same variation in community structure described above.

## Predicting recovery

Since the PERMANOVA test indicated that there was no significant difference in the meiofauna community structure between the Before community and four years after turbidity flow, this community can be considered recovered. To determine the recovery trajectory and when recovery may have occurred prior to the final sampling point, recovery rates for the meiofauna community were estimated using three different population growth

**Table 7  DISTLM results for the marginal and sequential tests for meiofauna and nematode community.** DISTLM results for the marginal and sequential tests for meiofauna and nematode community structure relationships with environmental variables before and after a turbidity flow in Kaikōura Canyon.

| Variable Meiofauna | Test Marginal | AIC | SS (trace) | Pseudo-F | P | Prop. |
|---|---|---|---|---|---|---|
| Depth (m) | – | | 800.13 | 1.002 | 0.365 | 0.046 |
| TOM% | – | | 2,363.60 | 3.264 | **0.037** | 0.135 |
| Chl *a* (µg/g) | – | | 653.15 | 0.811 | 0.437 | 0.037 |
| Chl *a*: Phaeo | – | | 625.69 | 0.775 | 0.468 | 0.036 |
| C: N (molar) | – | | 541.77 | 0.668 | 0.528 | 0.031 |
| PN% | – | | 2,137.20 | 2.908 | **0.049** | 0.122 |
| Skewness (F&W phi) | – | | 1,400.30 | 1.818 | 0.143 | 0.080 |
| <16 µm | – | | 1,771.50 | 2.355 | 0.085 | 0.101 |
| | Sequential | | | | | |
| TOM% | | 153.36 | 2,363.60 | 3.264 | **0.038** | 0.135 |
| Chl *a*: Phaeo | | 152.85 | 1,573.90 | 2.309 | 0.084 | 0.090 |
| Skewness (F&W phi) | | 152.82 | 1,153.30 | 1.756 | 0.155 | 0.066 |
| **Nematodes** | **Marginal** | | | | | |
| Depth (m) | – | | 1,533.00 | 0.563 | 0.913 | 0.086 |
| TOM% | – | | 3,561.90 | 1.493 | 0.132 | 0.199 |
| Chl *a* (µg/g) | – | | 4,149.30 | 1.813 | 0.073 | 0.232 |
| Chl *a*: Phaeo | – | | 1,914.60 | 0.720 | 0.740 | 0.107 |
| C: N (molar) | – | | 5,862.10 | 2.926 | **0.018** | 0.328 |
| PN % | – | | 2,827.20 | 1.127 | 0.304 | 0.158 |
| Skewness (F&W phi) | – | | 2,173.50 | 0.830 | 0.596 | 0.122 |
| <16 µm | – | | 3,575.30 | 1.500 | 0.129 | 0.200 |
| | Sequential | | | | | |
| C: N (molar) | | 62.518 | 5,862.1 | 2.926 | **0.016** | 0.328 |
| Chl *a* (µg/g) | | 62.245 | 2,973 | 1.643 | 0.063 | 0.166 |
| TOM% | | 61.589 | 2,555.6 | 1.575 | 0.124 | 0.143 |
| Chl a: Phaeo | | 60.595 | 2,026.4 | 1.362 | 0.301 | 0.113 |
| Skewness (F&W phi) | | 58.388 | 1,825.4 | 1.384 | 0.336 | 0.102 |
| PN % | | 56.28 | 1,059.6 | 0.671 | 0.605 | 0.059 |

**Notes.**
%TOM, percent total organic matter; Chl a, chlorophyll a concentration; Chl a: Phaeo, ratio for Chl a to phaeopigment; PhaeoC:N, organic carbon to nitrogen molar ratio; %PN, percent nitrogen; %TOC, percent total organic carbon; AIC, Akaike Information Criterion; SS, sum of squares; Pseudo-F, multivariate analogue Fisher's *F* test; P, *p*-value (significant values (<0.05) are in bold); Prop, indicates the proportion of variation explained by each variable.

models (linear, exponential, and logistic). These models confirmed the results from the PERMANOVA test and predicted that the impacted meiofauna community would exhibit the same within-group level of similarity as the pre-disturbance community (79%; the threshold used for predicted recovery) between 3.9 and 4.0 years after the turbidity flow (Fig. 5). The same three population growth models were used to estimate recovery rates for the nematode community, which the PERMANOVA test indicated was still significantly different 4 years after the turbidity flow. The models predicted that the impacted nematode

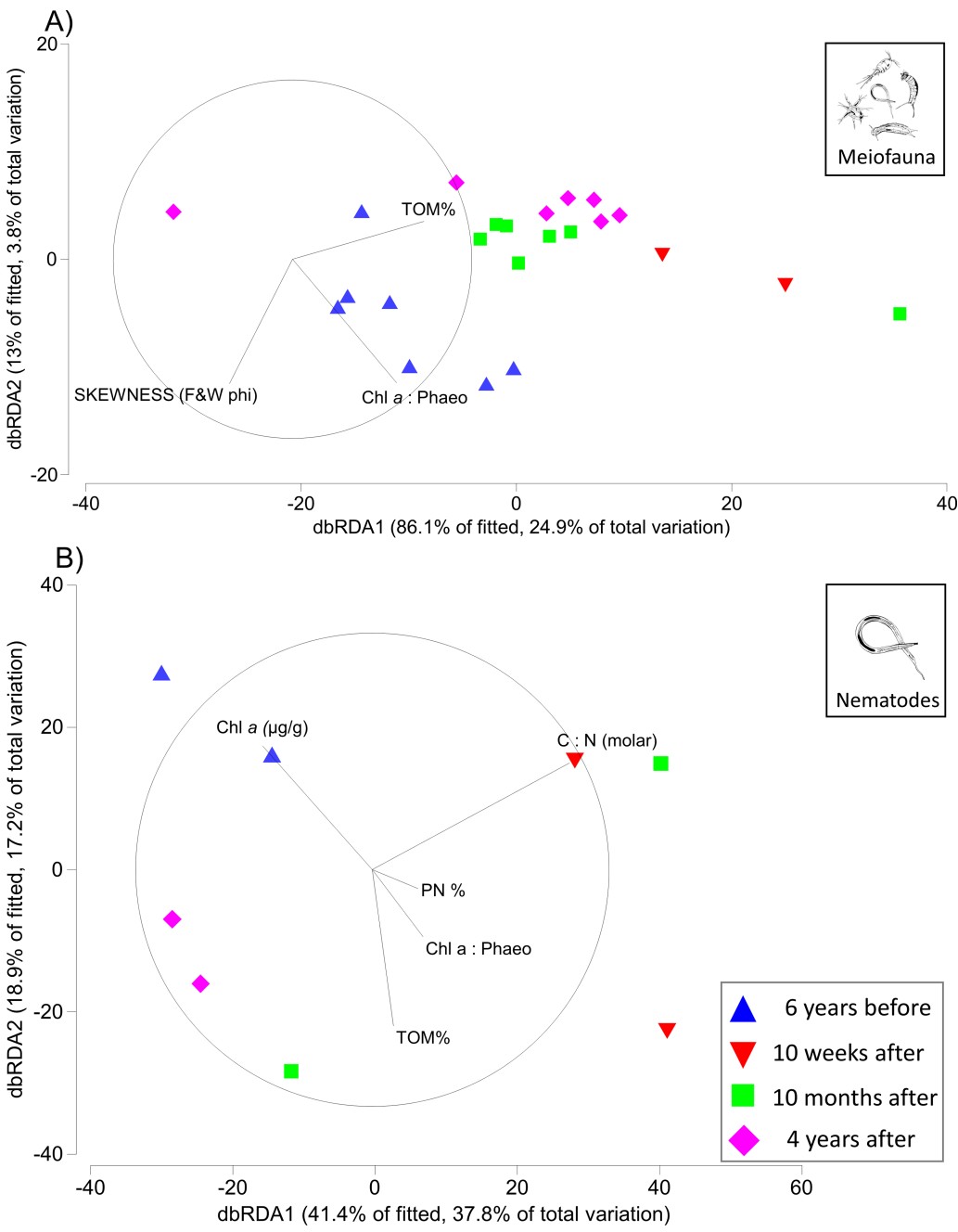

**Figure 3** **Distance-based redundancy analysis (dbRDA) plots for meiofauna and nematodes.** Distance-based redundancy analysis (dbRDA) plot visualising in two-dimensions the relationships between variation in community structure for (A) meiofauna and (B) nematodes (6 years before, and 10 weeks, 10 months, and 4 years after the turbidity flow event in Kaikōura Canyon) and environmental variables examined by the DISTLM analysis. Only variables with a Spearman rank correlation greater than 0.2 are displayed. Vector lengths are proportional to their contribution to the overall variation.

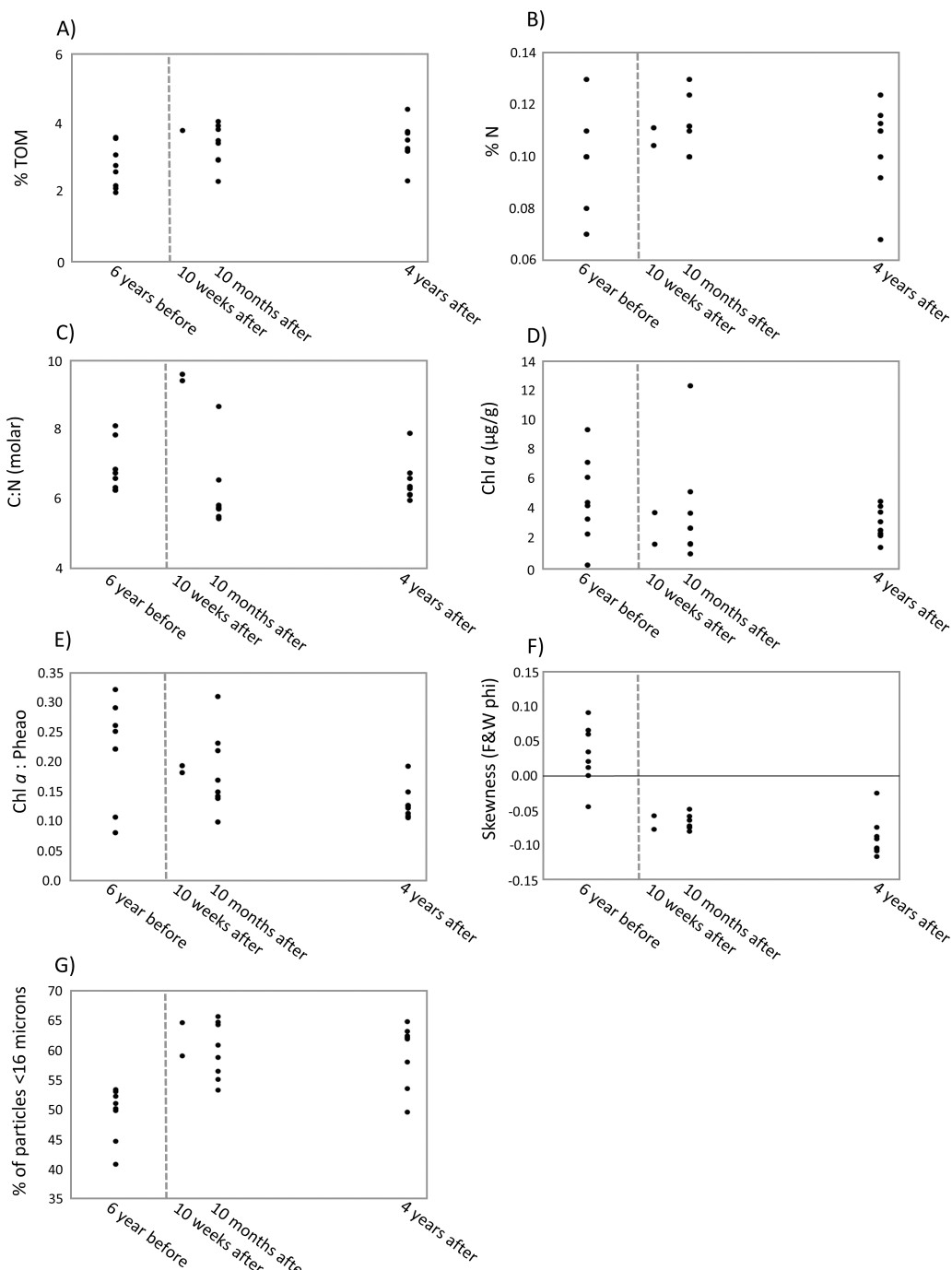

**Figure 4 Scatter plots of key environmental factors.** Scatter plots of the most important environmental factors identified by the DISTLM analysis for structuring meiofauna and nematode communities before and after a turbidity flow in Kaikōura Canyon. (A) The percent total organic matter (% TOM), (B) nitrogen (%N, C) the ratio of molar carbon (C) to nitrogen (N), D) Chl $a$ (mg g$^{-1}_{sediment}$), (E) ratio of Chl $a$ to phaeopigments, (F) the skewness of grain size, and G) the percent of grains less than 16 μm. Each dot represents a single core. The dashed line indicates when the turbidity flow in Kaikōura Canyon occurred.

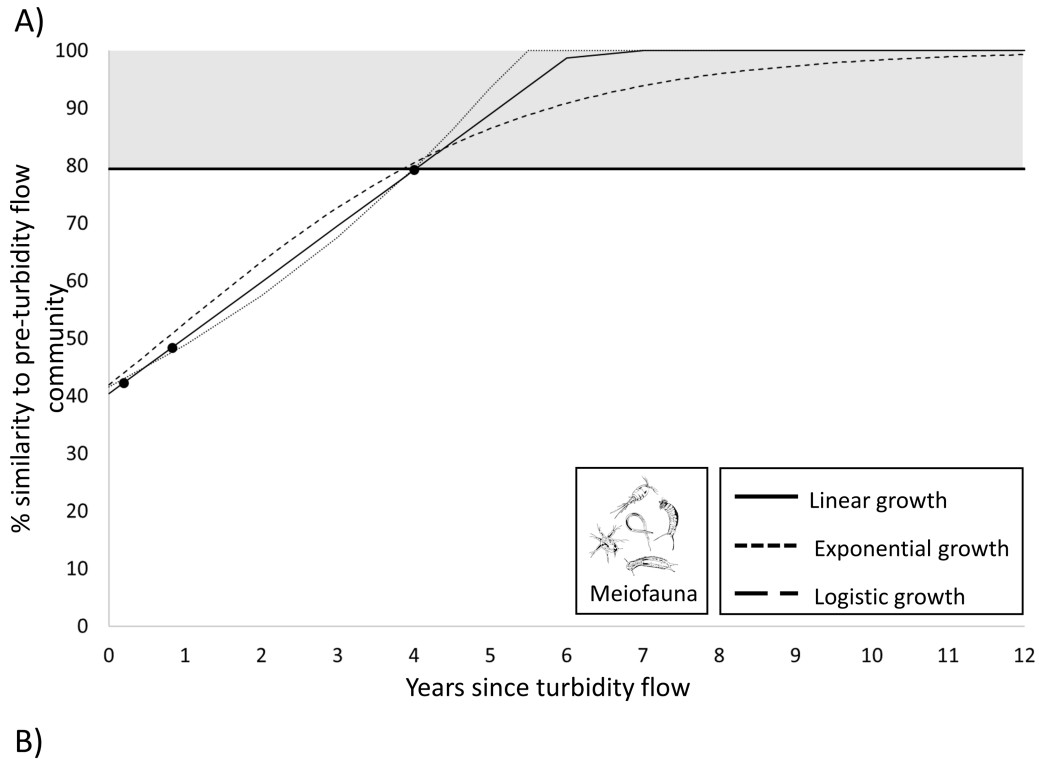

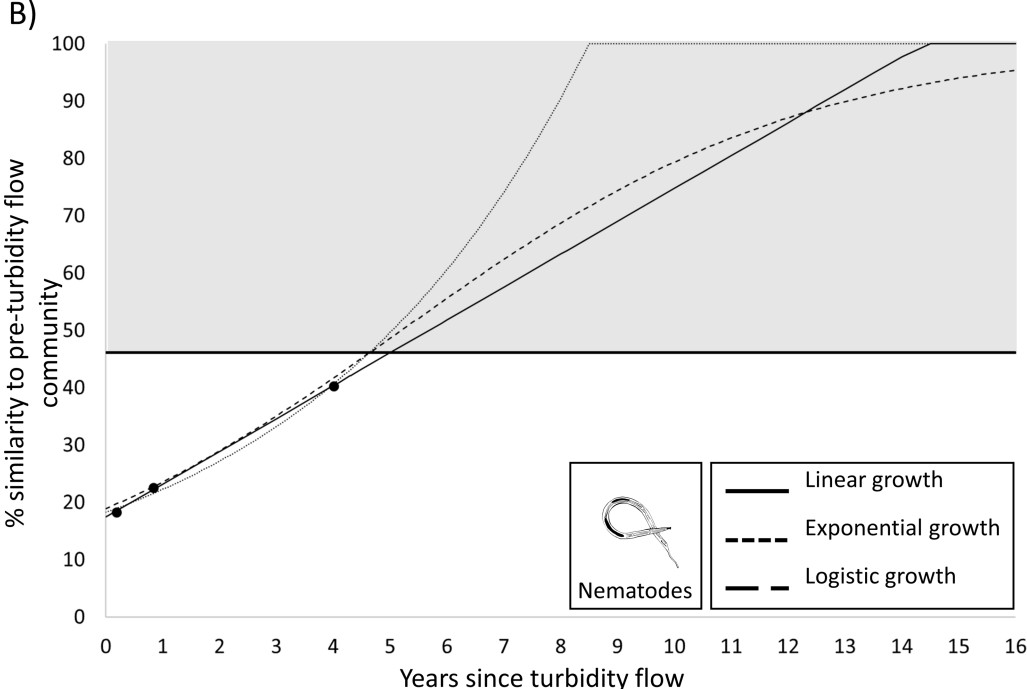

**Figure 5** **Plots predicting time to recovery for meiofauna and nematodes.** Plots showing three hypothetical models of population growth (linear, exponential, and logistic) used to predict the time to community recovery (indicated by the grey area on the plot; the minimum threshold of 79% or 46% similarity is the within-group similarity of the pre-turbidity community structure) for: (A) the meiofauna and (B) nematode communities in Kaikōura Canyon.

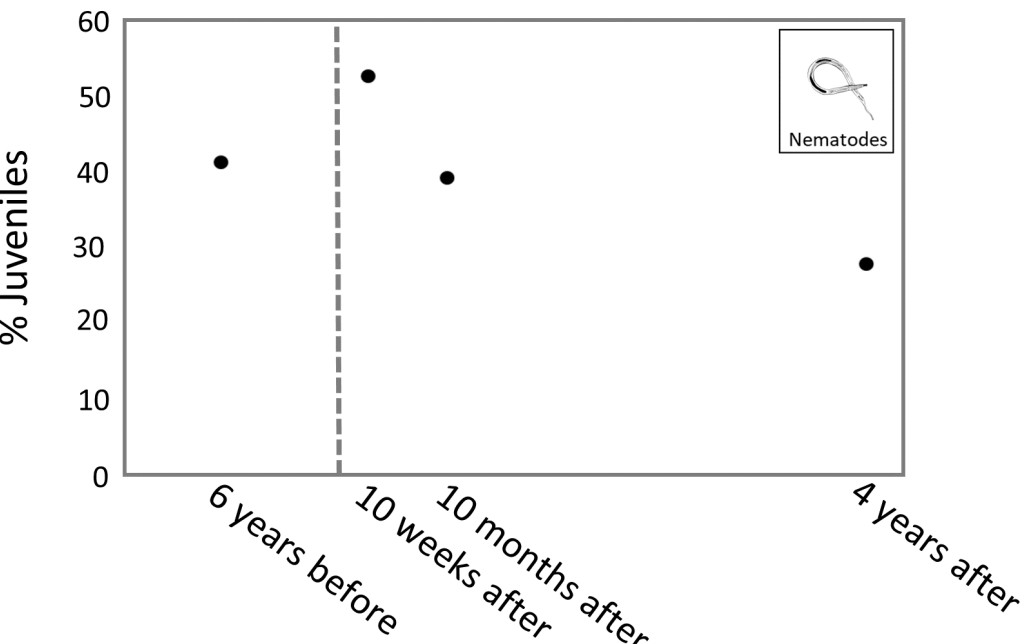

**Figure 6   Plot of juvenile nematode percentages through time.** Plot showing the average percentage of juvenile nematodes from sites K2 and K3 at each time point. The dashed line indicates when the turbidity flow in Kaikōura Canyon occurred.

community could exhibit the same within group level of similarity as the pre-disturbance community (46%) between 4.6 and 5.0 years after the turbidity flow.

### Nematode juvenile percentage

The highest percentage of juveniles was observed 10 weeks after the turbidity flow, at 53.1%. The percentage of juveniles at 10 months and 4 years after the disturbance was 39.6% and 28.1%, respectively. The percentage of juvenile nematodes 6 years before the turbidity flow was 41.6% (Fig. 6).

## DISCUSSION

### Impact of turbidity flow on meiofauna and nematode community structure

The meiofauna community sampled in Kaikōura Canyon was not resistant to disturbance caused by the 2016 Kaikōura Earthquake-triggered turbidity flow, but it appears that the community is resilient because by 4 years after the event the community had largely recovered. However, when considering the nematode component separately—the largest component of the meiofaunal community—using species level identification data (for a sub-set of the study sites), it appears that nematodes were still on a trajectory to recovery, as it had not yet recovered 4 years after turbidity flow disturbance.

It is evident that the meiofauna community was significantly altered by the disturbance with dissimilarity highest between Before and 10 weeks after the turbidity flow: meiofauna. The community was in a similar state 10 months after the event, though dissimilarity
between the community and the Before event community had decreased. The level of dissimilarity between Before the turbidity flow and 4 years after the event had decreased considerably for both meiofauna and nematode communities, and the meiofauna community was no longer significantly different from community sampled before the disturbance. However, while the nematode community was beginning to resemble the pre-disturbance community, there was still a significant difference in community structure. These findings were supported by the sample dispersion values for the meiofauna community which were highest after the disturbance but had returned to a level similar to pre-disturbance by 4 years after. While recovery is occurring the trajectory of the recovery is not comparable to a simple cyclical pattern, which assumes that roughly the same amount of recovery will occur between each time step, indicating another pattern may better describe the meiofauna community's recovery (see below).

The meiofauna community before the disturbance was dominated by nematodes (*Leduc et al., 2014*; *Leduc et al., 2020*; Fig. 7A). The key nematode species included *Hopperia beaglense, Cervonema kaikouraensis, Camplyaimus* sp. 6, *Leptolaimus* sp. 14, and *Sabatieria* sp. A. The difference in the community Before and 10 weeks after the disturbance event is characterised by a large decrease in abundance of all key taxa (Fig. 7B). For example, the abundance of *Sabatieria* sp. A decreased from 355 ind./10 cm$^2$ to 11 ind./10 cm$^2$ 10 months after the disturbance. This drastic abundance reduction in most taxa is to be expected given the evacuation of substrate from the canyon head, which would have removed most if not all of the living meiofauna community that resided within those sediments prior to the event. Similar removal of all or most fauna has been documented in other studies where substantial amounts of near-surface material are removed, such as harbour and aggregate dredging (*Kenny & Rees, 1994*; *Szymelfenig, Kotwicki & Graca, 2006*). From 10 weeks to 10 months after the turbidity flow, the abundance levels of most key taxa remained depressed compared to pre-disturbance levels (Fig. 7C). Some taxa saw minor increases from 10 weeks to 10 months after the turbidity flow, likely due to their recovery (*i.e., Sabatieria* sp. A, 10 weeks: 11 ind./10 cm$^2$; 10 months: 18 ind./10 cm$^2$). While other taxa decreased in abundance or were not seen at all 10 months after the disturbance. For example, kinorhynchs decreased from 2 ind./10 cm$^2$ to <1 ind./10 cm$^2$ and the nematodes *Camplyaimus* sp. 6 and *Leptolaimus* sp. 14 were not observed despite being present at 10 weeks after the turbidity flow (potentially due to fecundity levels, see below). Decreases in kinorhynch abundances have been reported following organic enrichment and associated increases in sulphide concentrations (*Mirto et al., 2012*; *Dal Zotto et al., 2016*). Alternatively, with such low abundances post-turbidity flow the missing taxa may have been present in the overall habitat but not sampled by the two cores analysed for this study. The exception to these small changes in abundance at 10 months after was *Daptonema* sp. 18 which increased to almost three times pre-disturbance abundances. Consistent with these Kaikōura observations, nematodes in the *Daptonema* genus are opportunistic, non-selective deposit feeders that are commonly found in disturbed, organic-rich sediment (*Vanreusel, 1990*; *Schratzberger & Jennings, 2002*; *Moreno et al., 2008*; *Liao, Wei & Yasuhara, 2020*).

Four years after the disturbance, the meiofauna community was no longer significantly different then the community before the disturbance, although not identical. Key taxa

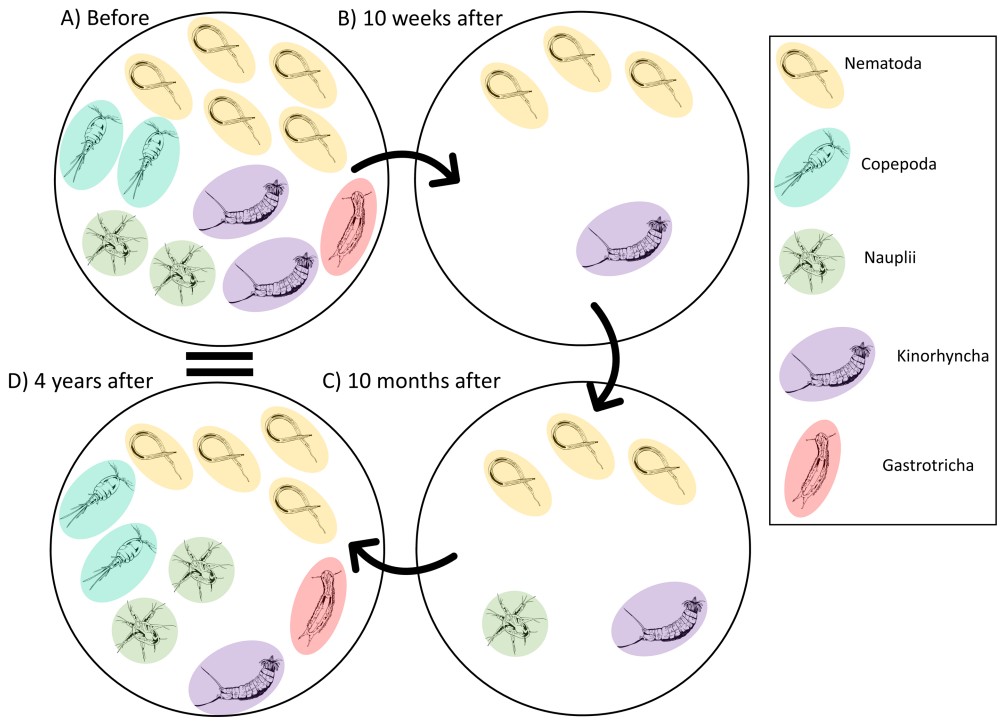

**Figure 7 Illustrated schematic showing the changes in the meiofauna community through time.** Schematic illustration showing of the relative abundances of the key taxa identified by the meiofauna SIMPER analysis that characterised the changes in the meiofauna community before and after the turbidity flow in Kaikōura Canyon. Solid arrows connect time points. One individual represents an average abundance of 1–10 ind./10 cm², two individuals represent an average abundance of 10–100 ind./10 cm², three individuals represent an average abundance of 100-1000 ind./10 cm², four individuals represent an average abundance of 1,000–2,000 ind./10 cm², and five individuals represents 2000+ ind./10 cm². Fauna illustration credit: Elise Littell.

in the meiofauna community at 4 years after were nematodes, which had recovered to near pre-disturbance abundance levels, nauplii and copepods, which were slightly more abundant than they had been before the disturbance, and kinorhynchs and gastrotrichs, which had much lower abundances than before the turbidity flow event (Fig. 7D). Copepods, gastrotrichs, and nauplii are considered to be more sensitive to stress than nematodes (*Murrell & Fleeger, 1989*; *De Troch et al., 2013*; *Pusceddu et al., 2013*; *Zeppilli et al., 2015*) providing some explanation for the longer time taken compared to nematodes to re-establish after the turbidity flow in Kaikōura Canyon. A similar relative abundance response by nematodes and copepods to a turbidity flow disturbance was observed after the Tōhoku Earthquake-triggered turbidity flow, where nematode densities remained unchanged after the disturbance, but harpacticoid copepod densities were negatively impacted by the disturbance and it wasn't until months to years after the event that they increased (*Kitahashi et al., 2014*; *Kitahashi et al., 2018*).

The percentage of juvenile nematodes peaked 10 weeks after the turbidity flow before decreasing 10 months and 4 years after. Conversely, meiofaunal annelids, which mostly

comprise juvenile polychaetes (*Warwick, 1988*), were least abundant 10 weeks after the turbidity flow but steadily increased in abundance at the 10 months and 4 years after time points. The increase in annelids over time after the disturbance indicates recruitment into the macroinfaunal community. The differences in juvenile abundance between these two groups is likely due to the differences in life histories. The peak of juvenile nematodes shortly after the disturbance suggests that their initial recruitment occurred primarily *via* juveniles rather than adults, probably due to the transport of juveniles from nearby unimpacted locations *via* sediment resuspension by currents (*Ptatscheck & Traunspurger, 2020*). Polychaete recruitment depends on the availability of larvae in the water column, which can be highly variable in time and space depending on reproductive cycles, abundance of adult populations, larval mortality and hydrodynamics (*Qian, 1999*).

The separate species level analysis of the nematode community provided, in particular, some additional understanding of the status of this important taxon 4 years after the turbidity flow, when this component of the meiofauna community had yet to fully recover. *Daptonema* sp. 18 dominated the community at the final sampling timepoint but with abundances 12 times higher than pre-disturbance levels. Other species such as *Sabatieria* sp. A were near pre-disturbance abundance levels, while *Camplyaimus* sp. 6 was observed but in very low abundances, and *Leptolaimus* sp. 14 was still not observed 4 years after the turbidity flow. Similarly, nematodes from the genus *Leptolaimus* were rare or absent in fresh iceberg disturbance scours in the Weddell Sea (*Lee et al., 2001*), despite otherwise often being a dominate taxa (*Vanaverbeke et al., 1997*; *Vanreusel et al., 2000*). This impact on these nematodes was attributed to relatively low fecundity of this genus (*Lee et al., 2001*). Two species that were not identified as key taxa before the turbidity flow, *Endeolophos* sp. 3 and *Microlaimus* sp. 34, were identified as key taxa 4 years after the disturbance. The genus *Microlaimus* makes up an important fraction of the nematode community in the Congo Channel, which is regularly disturbed by turbidity flows (*van Gaever et al., 2009*). The genus is considered to be an opportunistic coloniser and is often among the first taxon to recolonise physically disturbed patches (*e.g.*, *Lee et al., 2001*; *Raes, Rose & Vanreusel, 2010*).

Overall, 4 years after the turbidity flow disturbance the meiofauna and nematode community of Kaikōura Canyon has reattained pre-disturbance character of high abundance, low diversity, and dominance by a few species/taxa that are typically associated with high food and high levels of disturbance experienced in the canyon (*Leduc et al., 2014*). A similar meiofauna community pattern has been observed at other locations disturbed by turbidity flows (*Hess et al., 2005*; *Hess & Jorissen, 2009*; *Lambshead et al., 2001*; *Tsujimoto et al., 2020*; *van Gaever et al., 2009*) (see below). While the meiofauna community was not significantly different from the pre-disturbance community and could be considered recovered, the analysis of species level nematode data for two of the eight sites indicated that at this level the community was still significantly different from the pre-disturbance community, and therefore recovery was incomplete. Using the species level nematode data, recovery was predicted to occur between 4.6 and 5.0 years after the turbidity flow. These results indicate that while disturbance and community recovery can be detected using coarse taxonomic groups (*Warwick, 1988*; *Olsgard, Brattegard & Holthe, 2003*; *Musco et al.,*

*2011*), the use of species data give a more nuanced understanding of change and will likely indicate a longer recovery period than if a coarse taxonomic level is used. Predictions of recovery time suggest that a linear model may best describe the pattern of recovery the meiofauna exhibited, but this may be due to the limited number of repeated samples and additional points are necessary to help establish the recovery pattern.

## Comparison with other studies of turbidity flow disturbances

Other meiofauna communities impacted by turbidity flows have generally recovered rapidly from the disturbance. Overall, the meiofauna community impacted by the Tōhoku Earthquake-triggered turbidity flow recovered by 1.5 years after the disturbance (*Kitahashi et al., 2014*; *Kitahashi et al., 2016*). However, the foraminiferal component of the community was not yet considered recovered by this time (*Tsujimoto et al., 2020*), which contrasts with the foraminifera community of the Cap Breton Canyon which was considered recovered ∼1.5 years after a turbidity flow in this canyon (*Hess et al., 2005*; *Hess & Jorissen, 2009*). The difference between the recovery time of the Kaikōura Canyon meiofauna community and the Japan Trench slope community is notable since the Kaikōura Canyon sites are mostly in relatively shallower water depths (400–1,300 m) compared to the majority of the sites considered in the Tōhoku study (100–6,000 m), because it is generally held that organisms at deeper depths will take longer to recover from disturbances (*Nomaki et al., 2016*). The difference in these recovery times is likely due to scale of the disturbance at the locations. The Tōhoku Earthquake turbidity flow was less confined by seabed morphology and had a wider, but a reduced sedimentation impact on the Japan Trench slope (1–5 cm of deposition, 0.2 km$^3$ of transported sediment; (*Kitahashi et al., 2014*; *Kioka et al., 2019*) than the Kaikōura Earthquake turbidity flow had on Kaikōura Canyon (average erosion of 5.6 m, 0.9 km$^3$ of transported sediment; *Mountjoy et al., 2018*). Similarly, while the Cap Breton turbidity flow occurred in a canyon, it was considerably smaller (8–18 cm of deposition; *Anschutz et al., 2002*) than the Kaikōura Canyon turbidity flow.

Additionally, following the Tōhoku turbidity flow and the Cap Breton Canyon turbidity flow there was an apparent commensurate decrease in the distribution in the meiofauna community to the sediment subsurface (*Hess et al., 2005*; *Kitahashi et al., 2014*; *Nomaki et al., 2016*; *Tsujimoto et al., 2020*), potentially in response to burial of organic carbon or other structuring factors (see below). These vertical changes in distribution have also been observed in meiofauna communities from the Congo Canyon that have been impacted by turbidity flows (*Galéron et al., 2009*; *van Gaever et al., 2009*). Data from the present study do not have the same vertical resolution, because sediment slices were taken from 0–1 cm and 1–5 cm rather than one cm slices to five cm achieved for the Japan Trench slope samples. Hence, it is not possible to assess similar fine-scale changes in vertical distribution in the Kaikōura Canyon meiofauna following the turbidity flow. However, evidence from the megafauna and macrofauna components of the canyon community indicate that the overall community distribution did not change to be deeper in the substrate, and the distribution of some organisms may have instead changed towards the seafloor surface (*Bigham et al., 2023a*; *Bigham et al., 2023b*).

All three benthic size classes (mega-, macro-, and meiofauna) in Kaikōura Canyon were characterised by opportunistic species generally thought to be rapid colonisers and or those with traits that allow them to thrive in habitats with high food availability and high levels of disturbance. The estimated time to recovery for the meiofauna community in Kaikōura Canyon based on the coarse taxonomic level (3.9−4.0 years) is less than that predicted for both the megafauna (4.6−5.2 years; *Bigham et al., 2023a*) and macrofauna (5.6−6.7 years; *Bigham et al., 2023b*) communities. It has previously been hypothesised that meiofauna are more resilient to turbidity flow disturbances due to their rapid turnover times and lower sensitivity to changes in environmental factors (*Kitahashi et al., 2014*; *Kitahashi et al., 2016*; *Nomaki et al., 2016*). However, these new recovery estimates for the Kaikōura meiofauna may be an underestimate due to lower taxonomic resolution of these data (cf. *Smith & Simpson, 1993*; *Lasiak, 2003*; *Bates et al., 2007*) with a more complete recovery from the disturbance, as indicated by the nematode species level analysis, predicted to take longer (4.6–5 years), which is on par with the recovery estimate for the megafauna community (*Bigham et al., 2023a*) but faster than the macrofauna community (*Bigham et al., 2023b*).

## Changes in environmental factors and potential influences on the meiofauna community

The influence of environmental variables on community structure was modelled to provide further explanation for the pattern of meiofauna community structure observed in Kaikōura Canyon following the turbidity flow. The best model for describing the patterns of similarity observed in the meiofauna and nematode communities among the time points accounted for approximately 29% and 91% of the total variation, respectively. The amount of explanation for the meiofauna community is low, though not unusual for studies of deep-sea meiofauna communities (*e.g.*, *Zeppilli et al., 2013*; *Román et al., 2016*), but the amount of explanation for the nematode community is quite high, likely due to the relatively small dataset of only two sites and the species level taxonomic resolution of the nematode dataset.

The Kaikōura Canyon meiofauna community structure has previously been linked to high food availability in the canyon (*Leduc et al., 2014*; *Leduc et al., 2020*) and the findings from the environmental modelling in the present study suggest the same inference. The community structure over time post-event was best explained by the quantity and quality of the available organic matter and the skewness of the sediment, similar to the results for the macrofauna from Kaikōura Canyon (*Bigham et al., 2023b*). Post-turbidity flow, the organic matter content of the sediments increased but the overall quality of that organic matter decreased (as reflected in the decrease of Chl *a* to phaeopigment ratios in the sediment). The decrease observed in the concentrations of Chl *a* in the sediments (typically associated with the productivity of phytoplankton) after the turbidity flow, and the related change in the ratio of Chl *a* and phaeopigment concentrations, indicates that there was a decrease in the relative lability of the organic matter in the sediments. This change may have been due to the significant erosion caused by the canyon flushing event (*Mountjoy et al., 2018*) uncovering older, less labile organic matter or due to an increase in terrestrial material entering the canyon (*e.g.*, *Gibbs et al., 2020*) following landsliding
in the surrounding catchments and hinterland also triggered by the earthquake (*Dellow et al., 2017*; *Croissant et al., 2019*; *Massey et al., 2020*; *Thomsen et al., 2020*). The overall post-turbidity flow sediment particle size was negatively skewed reflecting an increase in finer particles. An increase in organic matter tends to be closely associated with an increase in fine sediments (*Keil et al, 1994*; *Mayer, 1994*; *Milliman, 1994*) so this change may simply reflect that increase in available organic matter, but it also reflects changes to the arrangement and structure of the physical environment. The physical environment has also been shown to drive changes in the fauna, particularly meiofauna which as the smallest size class that live in the interstitial spaces between sediment particles experiencing changes in the sediment matrix more strongly than larger fauna (*Tietjen, 1976*; *Heip, Vincx & Vranken, 1985*; *Etter & Grassle, 1992*; *Leduc et al., 2012a*).

The key variables identified by the environmental models for the nematode community are all connected to food quantity and quality. The most important variable was the ratio of C:N (molar). The relatively low C:N ratios inside the canyon 6 years before the disturbance was attributed to higher overall contributions of "fresh" marine organic matter (*Gibbs et al., 2020*). Ten weeks after the turbidity flow the C:N ratios were higher for both sites and were still high 10 months after the disturbance, which may be due, as noted above, to the canyon-flushing removing the fresher more labile organic matter and/or exposed older organic matter (*Okey, 1997*). By four years after the disturbance, the C:N ratio attained pre-disturbance levels for all sites, indicating that availability of labile organic matter had returned to pre-disturbance levels (*Gibbs et al., 2020*). Other important variables identified by the model were an increase in %TOM after the turbidity flow, a decrease in Chl *a* concentrations, and the Chl *a* to phaeopigment ratio in the sediments, and an increase in percent nitrogen in the sediment, reflecting an increase in food availability but a decrease in the quality of that food, as suggested by the relatively elevated C:N ratios (see also above for meiofauna community overall). Other, non-turbidity flow, studies have found correlations between deep-sea nematode density and distribution and food quality (*Levin, 1991*; *Neira et al., 2001*; *Gallucci et al., 2008*).

The relatively small amount of variation in community structure explained by the model for the meiofauna may be because of the coarse taxonomic resolution of the data, as evidenced by the higher variation explained by the higher resolution nematode data, or because of other unmeasured biological or environmental factors are instead mainly responsible for the recovery process. For example, the oxygenation and chemical conditions of the sediments have been hypothesized and found to structure meiofauna communities after other turbidity flows. Though not measured sediment oxygen levels were postulated as a driving factor for meiofauna communities impacted by turbidity flows in Cap Breton Canyon (*Anschutz et al., 2002*; *Hess et al., 2005*; *Hess & Jorissen, 2009*). A study of meiofauna following the Tōhoku Earthquake-triggered turbidity flow found that sediment oxygen levels were a key structuring factor for meiofauna-sized copepods (*Nomaki et al., 2016*). Additionally, oxygen limitation has been proposed more broadly as a direct control on deep-sea meiofauna composition at higher taxonomic levels (*e.g.*, copepods and nauplii density; *Levin, 1991*; *Neira et al., 2001*). In Kaikoura Canyon, a study of sediment mixing depth from 4 years after the disturbance found that the maximum

mixing depth was 2.19 cm, which may be mediating sediment oxygenation in the canyon (*Hale et al., 2024*).

## Management implications

Kaikōura Canyon was designated part of the Hikurangi Marine Reserve in 2014 because it is a benthic productivity hotspot (*De Leo et al., 2010*) and provides wider ecosystem services (*Fernandez-Arcaya et al., 2013*), including hosting an abundant marine mammal and avifauna (*e.g.*, *Guerra et al., 2020*). Concerns were raised following the 2016 Kaikōura Earthquake-triggered canyon flushing event that the efficacy of the reserve had been impacted. Results from this study show that overall, the meiofauna community had largely recovered 4 years after the turbidity flow. However, a more complete recovery from the disturbance, as indicated by the nematode species level analysis, was predicted to take longer (a minimum of 4.6–5 years, *i.e.,* somewhere between 2023 and 2024) and additional samples are necessary to test this prediction. These additional samples would better establish the shape of recovery trajectory patterns and to see if recovery time falls higher on the curve and closer to maximum predicted time to recover.

Natural disturbances in the deep sea have been considered as potential proxies for anthropogenic disturbance with varying levels of validity (*Angel & Rice, 1996*; *Tyler, 2003*). Debris and turbidity flows create large-scale erosional and depositional disturbances, and thus, could be considered as proxies for some anthropogenic disturbances, such as deep-sea seabed mining where extraction and dredging/turnover of the seafloor can occur. However, results from *Bigham et al. (2023a)*, *Bigham et al. (2023b)* on the recovery of the megafauna and macrofauna component of the Kaikōura Canyon benthic community suggest that the impacts of a turbidity flow on a benthic community was not readily transferable to understanding the impact of future deep-sea mining. This conclusion appears to also be the case for the meiofauna component of the Kaikōura Canyon benthic community. Studies of the impact of small-scale experimental deep-sea mining-related disturbances on meiofauna have shown that fauna at abyssal sites have not recovered to baseline levels after decades (*Miljutin et al., 2011*; *de Jonge et al., 2020*). In contrast, this study estimates that the Kaikōura Canyon nematode community structure could be recovered as soon as 4 years (meiofauna) and 4.6 years (nematodes) after the disturbance, although recovery could take up to 8 years or longer if different levels of community similarity were used as the threshold for recovery. The discrepancy in recovery timing and general lack of transferability between this natural disturbance and seabed mining is likely due to meiofauna communities in Kaikōura Canyon being subjected to much higher levels of natural disturbance from submarine landslides and turbidity flows than abyssal plains where mining for polymetallic nodules may occur in the future. Furthermore, as discussed above, the meiofauna in the canyon are likely to be more adapted to be resilient to these large-scale disturbances. For example, the genera and species of nematodes within Kaikōura Canyon are atypical of deep-sea nematode communities and instead are typically associated with high food availability and high disturbance levels (*Leduc et al., 2012*; *Leduc et al., 2014*; *Leduc et al., 2020*). Further, there are discrepancies in the habitat type as well as the scale of the disturbances. In the case of polymetallic nodules, the nodules themselves

constitute a unique habitat with meiofauna communities living on and in them that are distinct from the surrounding soft sediments, and which would be predominantly removed by the mining (*Thiel et al., 1993*; *Bussau, Schriever & Thiel, 1995*; *Veillette et al., 2007a*; *Veillette et al., 2007b*). In contrast the habitat on the floor of Kaikōura Canyon is a mostly uniform soft sediment. Erosion and deposition of sediment by the canyon-flushing event in Kaikōura Canyon was on the scale of metres to tens of metres (*Mountjoy et al., 2018*), much greater than the tens of centimetres to metres of erosion (*Levin et al., 2009*)and millimetres of deposition (*Thiel et al., 2001*) that are expected to occur from seabed mining. The minimum areal extent of the impact from the turbidity flow in Kaikōura Canyon was approximately 220 km$^2$ (*Mountjoy et al., 2018*), which although it is comparable to the hundreds km$^2$ per year impacted area envisaged for manganese nodule mining in the abyss (*Ardron et al., 2019*) seabed mining is expected to occur over successive and multiple years, and therefore may ultimately extend hundreds to thousands of square kilometres (*Smith et al., 2008*). As such, the recovery estimates from the Kaikōura Canyon study of the impact of turbidity flows on benthic communities are not likely to be good proxies for the recovery of such communities from deep-sea mining on abyssal plains.

## CONCLUSIONS

The meiofauna community, identified at a coarse taxonomic level, sampled from sediment cores from Kaikōura Canyon appears to be a resilient to the earthquake-triggered turbidity flow and has apparently recovered 4 years after the event. However, analysis of species level nematode data indicates that this component of the community had not yet recovered by this timepoint and is predicted to take a minimum of 4.6 years to recover. Future sampling at the same sites remains key to ascertain if or when the meiofaunal communities will fully recover. The pattern of resilience for the meiofauna community is somewhat in contrast to those for the megafauna and macrofauna communities examined in previous studies (*Bigham et al., 2023a*; *Bigham et al., 2023b*). With data from all three size classes available from Kaikōura Canyon it is now possible to synthesize the overall community resilience and examine inter-size class interaction dynamics during recovery.

## ACKNOWLEDGEMENTS

We would like to thank the captain, crew, and scientific parties of the R/V Tangaroa voyages TAN0616, TAN1006, TAN1701, TAN1708, and TAN2011, Grace Frontin-Rollet (NIWA) for direction on processing sediment samples, and Katherine Maier (NIWA) for discussions and insights on sediment particle size data. Meiofauna illustrations in figures were drawn by Elise Littell.

### Funding

Funding for this project came from NOAA Ocean Exploration and NIWA, with co-funding from Woods Hole Oceanographic Institution, Scripps Oceanographic Institution, and the University of Hawaii (TAN0616), predecessors of the present New Zealand Government research funding agency, the Ministry of Business, Innovation and Employment, *via* the Ocean Ecosystems programme (TAN1006), New Zealand Ministry for Primary Industries with additional funding from NIWA Strategic Science Investment Fund (SSIF) project COES1701 (TAN1701), NIWA SSIF and Tangaroa Reference Group (TRG) (TAN1707), NIWA's Coast & Oceans Centre and TRG (TAN1708), NIWA SSIF program COPR, TRG, and Eurofleets+ (TAN2011). Katharine T. Bigham was supported by a NIWA-VUW PhD scholarship in marine sciences. The funders had no role in study design, data collection and analysis, decision to publish, or preparation of the manuscript.

### Grant Disclosures

The following grant information was disclosed by the authors:
NOAA Ocean Exploration and NIWA.
Woods Hole Oceanographic Institution.
Scripps Oceanographic Institution.
University of Hawaii: TAN0616.
New Zealand Government Research Funding Agency.
Ministry of Business, Innovation and Employment: TAN1006.
NIWA Strategic Science Investment Fund (SSIF): COES1701 (TAN1701).
NIWA's Coast & Oceans Centre and TRG: TAN1708.
NIWA SSIF program COPR, TRG, and Eurofleets: TAN2011.
NIWA-VUW PhD scholarship.

### Competing Interests

The authors declare there are no competing interests.

### Author Contributions

- Katharine T. Bigham conceived and designed the experiments, performed the experiments, analyzed the data, prepared figures and/or tables, authored or reviewed drafts of the article, and approved the final draft.
- Daniel Leduc conceived and designed the experiments, authored or reviewed drafts of the article, supervision, and approved the final draft.
- Ashley A. Rowden conceived and designed the experiments, authored or reviewed drafts of the article, supervision, and approved the final draft.
- David A. Bowden conceived and designed the experiments, authored or reviewed drafts of the article, supervision, and approved the final draft.
- Scott D. Nodder conceived and designed the experiments, authored or reviewed drafts of the article, leading data collection, data interpretation, and approved the final draft.

- Alan R. Orpin conceived and designed the experiments, authored or reviewed drafts of the article, leading data collection, data interpretation, and approved the final draft.

## Field Study Permissions

The following information was supplied relating to field study approvals (i.e., approving body and any reference numbers):

Field sampling was undertaken under the General Special Permit issued by Fisheries New Zealand to the National Institute of Water and Atmospheric Research

## Data Availability

The meiofauna abundances and environmental variables are available in the Supplementary Files.

## Supplemental Information

Supplemental information for this article can be found online at http://dx.doi.org/10.7717/peerj.17367#supplemental-information.

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
