# Peer review of "Recovery of deep-sea meiofauna community in Kaikōura Canyon following an earthquake-triggered turbidity flow"

_PeerJ, doi:10.7717/peerj.17367_

## Round 0.1 · original submission · Minor Revisions

Dear Authors,

Thank you very much for submitting to PeerJ. I personally found the manuscript very interesting and innovative for the topic. For what I read from the reviewers, even they found the manuscript well-written, clear, and informative. Notwithstanding, I would consider the manuscript as a starting point to investigate in more detail events like the one described since the sampling design (apart the time span of sampling) could be ameliorated to consider several other aspects (e.g., there is not a comparison with the meiofauna communities in the grounds above the canyon to search for evidence of a shift in the community due to contamination from upper sediments). Like reviewer 1, I found the management part a little forced when comparing the topic of the manuscript with seabed mining as they seem two problems of different order of magnitude (see my comments below). I like the “review” cut you gave to the article even if I think that this weighs down a bit 'reading because they are given a lot of information. If it is possible, try to shorten the text avoiding repetitions or, in alternative, consider modifying the text as a real review (I suppose this would mean adding some more information, citations and possibly comparisons with other studies like this). This is just a suggestion so feel free to consider this modification. Finally, I agree with the reviewers that your manuscript requires only minor revisions but read deeply their comments and accomplish all their requests.

The introduction explains well the background of the work and I have no comments on this.

Material and methods
Line 127: Figure 1, can you enlarge the characters in the map? K01, K02, and so on. Change colour of the distance bar into white and bold it to make it more readable. Put a north arrow. Report also the mining of the K labels in the caption.
Line 148: Figure 1C, I can’t see any figure 1C.
Line 155: Only one core? Replicated in someway?
Line 161: Why 45 and not for example a 30 microns to retain even smaller individuals that possibly could react differently than bigger to the sediment flow?
Line 167: How many nematodes did you catch for identification from each replicate? Did you identify all the individuals, or you subsampled them (e.g., 100 individuals from each replicate)?
Lines 182-184: At least report the names of the analysis and then cite Bigham for details.
Lines 186-203: All this part doesn’t fit the statistical analysis paragraph. Please, move it in an appropriate paragraph, e.g., in the subsequent community structure, avoiding repetitions.
Line 208: with no replicates? Please, report also this information.
Lines 216-218: “…analysis replicate cores…” It is not clear the use of the term replicates. Please, explicit better if you collected (and how many) replicates you collected in the appropriate paragraph. Moreover, explain better the analysis strategies adopted for meiofauna and nematode. Why they were different? Why the meiofauna replicates were averaged and the nematode ones not?
Line 233: If you want to report the extended name of the procedure, please, report it the first time it appears. For consistency, report the name of all the procedures adopted or none.
Line 240: Again, just a table to have an idea of what those environmental variables are could be useful to the lazy reader that do not want to read another paper.
Lines 212, 237, 252: all these are subparagraphs of the statistical analysis but in this format version they are almost confusing as they seems as paragraphs per se. So, delete and rephrase them opportunely.
Lines 258-259: Report the functions used and the version of the packages. Moreover, report also model selection followed and the way to assess the models (e.g., AIC, BIC,…)

Results
Line 294: Table 6, I don’t understand this jump from table 3 to table 6. What about tables 4 and 5?
Line 330: PN%, I would consider this variable with caution as it has a border line p value of 0.049. Please, report this caution in the text.
Discussion
Lines 384-387: what about the nematode community from shallower depth transported by the flow to the canyon? Could them affect the previous communities after 4 years? You would not observe this shift when considering meiofauna communities maybe because a nematode is a nematode when not classified at genus or species levels, but conversely, when analysed more in details, the species could describe a completely different story. Did you check similarity with low depth communities? This could be a to do action for future analysis on this topic.
Line 609: According to reviewer 1 this part doesn’t sound well. The problems of sediment flows and seabed mining are two different order of magnitude problems. In the first case we have an external sediment flow, “exogenous” if you want (in the sense that the sediment comes from the above levels with possible community contamination from other shallower communities), in the second case we have a, “endogenous” resuspension of sediments with same communities only mixed. I suggest the authors to reconsider this part and concentrate in the already substantial discussion of turbidity flow in deep see sediments.

·

Basic reporting

The paper by Bigham et al. provides a compelling description and discussion of the KaikMura Earthquake, the resulting canyon-flushing event in KaikMura Canyon and the effects on meiobenthic communities and nematofauna in terms of diversity and recovery time. The manuscript is clear, well written and supported with adequate literature references, figures and tables. Overall, Results are relevant to the main goal of this research.

However, Some minor changes are needed before publication in PeerJ. Please, find my detailled comments and suggestions in the attached pdf.

Experimental design

Methods are clearly explained and adequate to accomplish with the main aim of the paper. A clarification is needed regarding the number of subreplicates for the meiofauna and the statistical analysis (see specifics in the pdf).

Validity of the findings

Results are well presented and statistically robust. The numbering of the Tables must be checked. Discussion is clear and well supported by literature references but can be shorten in certail parts.
The only weakness related to this part is the Management implications section. Authors wrote a long paragraph to conclude that their study cannot be used to evaluate mining impact. I would suggest to develop this part with other arguments (see my suggestions at lines 673-675) and maybe include it in the Conclusions or delete it.

Additional comments

no other comments to add.

Reviewer 2 ·

Basic reporting

The manuscript is well-written and organized. It adheres to the PeerJ manuscript structure guidelines, and its research subject is appropriate for the journal's aquatic ecology thematic area. Indeed, meiofauna is a challenging subject to investigate since there is a significant gap in our understanding of that disparate group of organisms, owing to their small size, incredible diversity, and taxonomic complexity. This group is highly useful in assessing the environmental impacts caused by both naturally occurring and anthropogenic, large seafloor disturbances, and the management implications brought up by the authors resulted very incumbent. However, authors warned that recovery estimates in Kaikōura Canyon are not likely to be good proxies for meiofaunal communities subjected to disturbances due to mining activities on abyssal seafloor. The increasing pressure on seabed mineral resources, which increases exploitation mining activities in many submarine areas in the world oceans, underscores the current importance of furthering knowledge in how marine biota responds to large disturbances, as the one explored in the revised manuscript.

Experimental design

The authors did mention that they do not include details on statistical methods employed in their manuscript because those were extensively explained on the two previous published papers on the macro and megafauna components of the same project. However, I think that is an unnecessary statement since the statistical tests and routines used in the current manuscript were well-explained and sufficient for readers to understand how they analyzed their results. If the authors still consider it necessary to mention it, then they should clearly state what those details are that are not fully developed in the section on “Statistical Analysis”.

I would like to have a better explanation, here in the methods section and in the results and discussion sections, on the test of cyclicity and how that metric informs on and discard that “the meiofauna community’s pattern of recovery was not comparable to a simple, equal distance cyclical recovery” (Ln281-282).

I would like to see a measure of uncertainty associated to the three predictive models for meiofauna and nematode datasets. At least include confidence intervals along the generated graphical function in the plots.

Validity of the findings

With respect to the predictive analyses conducted to estimate recovery times of meiofauna and nematodes, through adjusting to three different models, I acknowledge that the authors mentioned on lines 493-494 about needing additional points and a higher replication. However, I would invite authors to include the confidence intervals along those lines, representing the recovery trajectories for both the meiofauna and the nematode communities’ models. That is necessary as a measure of uncertainty for the predictions that those models are proving.

The analysis on how nematode community changes as time progresses after the turbidity flow event requires more emphasis on the ecology of the nematode species. I think that authors should exploit and take advantage of the high taxonomic resolution reached with the nematode dataset. May the authors elaborate more on the potential reasons why did some species appear to be recovering disproportionally after the event and others did not? And why did some species disappear during the recovery trajectory? Also, relationships between basal resources, such as OM and sediment chl-a, and species identity in the different communities studied between time points could be helpful. Perhaps looking at feeding groups among the nematode species could help to inform on patterns of abundance changes between time points, as environmental variables change.

Additional comments

Ln261-263; at the end of Materials & Methods, the sentence “The meiofauna and nematode community were predicted to be recovered when they at least reached the level of within-group similarity exhibited by the pre-turbidity flow community (i.e., 79% and 46.1%, respectively)” belongs to the Results section.

Ln268-269; the reported p-value in Table 2 for the “before-10 months after” pair-wise comparison is 0.0002, and not less than 0.0001 as appears in the text.

Ln287-288; range reported for the nematodes contribution to community similarity should say “between 66 and 77 %” to be coherent with Table 3.

Ln288-290; in the following sentence “with copepods also being key contributors 10 months after the turbidity flow and nauplii key contributors 10 weeks after the disturbance event (Table 3).” should say that copepods were key contributors 10 weeks after and nauplii 4 years after to be coherent with Table 3.

Ln304-306; The three main group contributors to dissimilarity between before and four years after were nematodes, nauplii, and copepods (according to Table 5) and not nauplii, nematodes, and gastrotrichs (as appears in the narrative).

Ln309; The table to be called at the end of the paragraph should be Table 5 and not Table 6.

Ln-295-309; In this paragraph, authors did not describe the last part of Table 5. At describing results corresponding to the main groups responsible for meiofauna community dissimilarity between time points (from Table 5), authors did describe comparisons between the before and after the event meiofauna communities but skipped comparisons among the three samplings made after the event, i.e., 10 weeks-10 months, 10 weeks-4 years, and 10 months - 4 years.

Ln310-327; same observation made with paragraph between line 295-309. Authors did not describe how was the community progression (this time nematode community data) in terms of dissimilarity among samplings after the event.

Ln341; the second dbRDA axis in the nematode plot explains 17.2% and not 17.9%, according to figure 3b. Please, correct it in the narrative.

Ln357-358; The last sentence “Additionally, highly correlated but removed variables would likely also explain the same variation in community structure described above.” is confusing and does not contribute to the main idea of the paragraph.

Ln388-430; Large parts of this extensive paragraph should be placed in the results section. In fact, the missing descriptions from tables 5 and 6 are being detailed here in the discussion. It makes the discussion unnecessarily long, filled with narrative that corresponds to results.

Ln522; In the discussion authors mentioned that sediment slices were taken “from 0-1 cm and 1-5 cm”, but in the Methods section they stated that for this manuscript they analyzed data pooling in both core slices and reported abundances from 0 to 5 cm. This is confusing.

On Table 5, the word gastrotrich is misspelled.

On Table 7, please identify all abbreviations used to name environmental variables in the caption or as foot notes below the table.

Figure 2. This compound figure may be better represented with only two panels, one for meiofauna and the other for nematodes. Just need to represent the centroids and the individual points in the same panel (only one panel for meiofauna and one for nematodes). I do not see that by doing this change, the figure will lose clarity or become cluttered. Instead, I think that is a good way to gain space. Also, I think that it is more appropriate to draw centroids in the same ordination plot where the original points for each group are represented and not to run an extra NMDS with just the centroids (which will produce plots with different stress values, for example). Another observation is that the centroid for the meiofauna group 10 years before the event is missing. Finally, the last sentence in the figure’s caption “All stress values are below 0.2, indicating that the plots are acceptable representations of the similarity patterns.” is not needed here. It is better to mention that in the narrative of the results section.

Figure 4. Please identify the dashed line as it was in Figure 6 panels: “The dashed line indicates when the turbidity flow in Kaikōura Canyon occurred.”

---

## Round 0.2 · accepted · Accept

Dear authors, I congratulate you, the manuscript is improved after the reviewers' and my personal suggestions. Only one reviewer accepted to revise the manuscript but I am personally satisfied with all your responses.

·

Basic reporting

Overall, I found the manuscript improved and I appreciated the effort made by the authors in taking in cosideration all comments and suggestions made by the Reviewers.

Unfortunately, I am still not convinced about this point: "studying the impacts of turbidity flows could be useful proxy for understanding seabed mining impacts on be benthic fauna was one of the motivations for the study".
The impacts, Canyon-flushing event and deep-sea mining, are very different kind of impacts even if both include turbidity effects.

At this point, I will leave the final decision to the Editor. I do not have any further comments and again I would like to stress the high quality of the MS.

Experimental design

No comments to add

Validity of the findings

No comments to add

Additional comments

No comments to add